# InvisibleInk: High-Utility and Low-Cost Text Generation with Differential Privacy

**Vishnu Vinod**[1]  **Krishna Pillutla**[1,2]  **Abhradeep Thakurta**[3]

[1]CeRAI, IIT Madras   [2]WSAI, IIT Madras   [3]Google DeepMind

## Abstract

As major progress in LLM-based long-form text generation enables paradigms such as retrieval-augmented generation (RAG) and inference-time scaling, safely incorporating private information into the generation remains a critical open question. We present INVISIBLEINK, a highly scalable long-form text generation framework satisfying rigorous differential privacy guarantees with respect to the sensitive reference texts. It interprets sampling from the LLM's next-token-distribution as the exponential mechanism over the LLM logits with two innovations. First, we reduce the privacy cost by isolating and clipping only the sensitive information in the model logits (relative to the public logits). Second, we improve text quality by sampling without any privacy cost from a small superset of the top-$k$ private tokens. Empirical evaluations demonstrate a consistent $8\times$ (or more) reduction in computation cost over state-of-the-art baselines to generate long-form private text of the same utility across privacy levels. INVISIBLEINK is able to generate, for the first time, high-quality private long-form text at less than $4$-$8\times$ times the computation cost of non-private generation, paving the way for its practical use. We open-source a pip-installable Python package (invink) for InvisibleInk at https://github.com/cerai-iitm/invisibleink.

## 1   Introduction

Large Language Models (LLMs) have demonstrated remarkable inference-time capabilities, synthesizing information from multiple sources into coherent responses via retrieval-augmented generation (RAG), and generating increasingly long and sophisticated outputs via the recent "deep research" and inference-time scaling paradigms [1–3]. Larger token budgets also allow these models to effectively "think," enabling self-reflection and correction of potential errors during generation [4–8].

The power of these models, however, often relies on their ability to process and reason over vast amounts of information, including potentially sensitive data provided at inference time. This context might come from user prompts containing private details or documents retrieved by RAG systems from confidential knowledge bases. There is some risk that the model's outputs, particularly in detailed long-form generation [9], could inadvertently leak sensitive details from this private context [10].

Differential Privacy (DP) [11] can mitigate such risks by providing provable guarantees that the output distribution is statistically indistinguishable whether or not any single piece of sensitive information was included in the input context. This guarantee is not just desirable but often essential in sensitive domains like healthcare and finance, where information leakage carries severe regulatory (e.g., HIPAA, GDPR) and ethical consequences and merely anonymizing personally identifiable information (PII) is not sufficient [12] Therefore, developing effective DP mechanisms for inference-time generation is critical for the responsible application of LLMs in privacy-sensitive applications. We tackle the challenge of achieving high-fidelity, long-form text generation under the rigorous DP guarantees on sensitive source text(s) used as references for generation.

39th Conference on Neural Information Processing Systems (NeurIPS 2025).

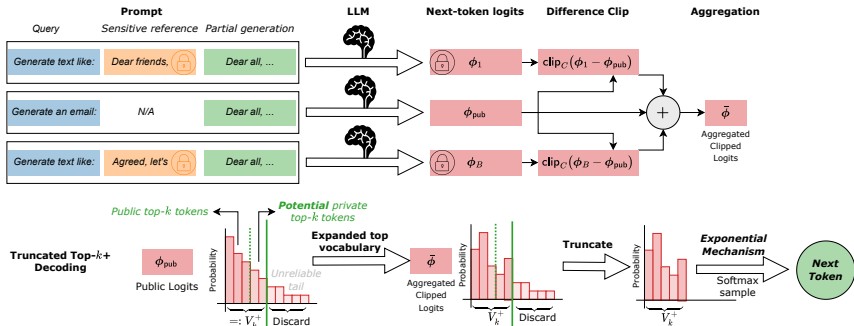

Figure 1: INVISIBLEINK interprets differentially private text generation as an iterative application of the exponential mechanism over a subset of the LLM's clipped logits. Our key innovations are: (a) DClip, an improved clipping function to reduce the sensitivity, and hence, the privacy cost; and (b) Top-$k$+ sampling, a truncated decoding algorithm to improve utility by selecting a subset of logits to sample each token from.

Despite the importance of this problem, existing approaches for DP text generation suffer from severe practical limitations. Consider the prior state-of-the-art method of Amin et al. [13], which interprets token sampling from the LLM as an instance of the canonical exponential mechanism [14] over the next-token logits. It requires a computational overhead of $100\times$ that of non-private generation or more, to produce non-degenerate text.[1] This makes them intractable at scale for large models.

Moreover, from a quality perspective, these methods typically resort to high-temperature sampling from the full vocabulary distribution. This decoding strategy is well known to produce degenerate or low-quality text, falling significantly short of the fluency and coherence achieved by standard non-private decoding algorithms, which rely on sampling from truncated distributions [16–18].

We bridge this gap with the following:

*We propose* INVISIBLEINK *for DP synthetic text generation. It can produce hundreds of tokens at* $8\times$ *lower computational cost at the same privacy/utility trade-offs compared to prior SoTA methods.*

Our main contributions are as follows:

- **INVISIBLEINK Framework**: We propose INVISIBLEINK, an exponential mechanism-based [14] sampling framework to allow LLM-based text generation with rigorous DP guarantees under the replace-by-null adjacency. We make two key innovations: (a) **DClip** to isolate and clip only the sensitive information in the model logits and avoid paying a privacy cost on prior information on language; and (b) **Top-$k$+ decoding**: a truncated decoding algorithm that approximates sampling from the top-$k$ private tokens with a *small* superset of size $k' \approx k$, ensuring high utility without losing out on private tokens that are unlikely under a public model.

- **Privacy-Utility-Compute Tradeoffs**: By empirically analyzing synthetic text generation on three domains—medical, legal, and commercial—we demonstrate an $8\text{-}16\times$ reduction in computation cost across the board to achieve the same privacy and utility.[2] Thus, INVISIBLEINK can produce hundreds of tokens at significantly smaller compute budgets (such as $< 10\times$ the non-private cost), including in settings where baselines extinguish their privacy budgets within a handful of tokens.

- **User-friendly Accounting**: Given a privacy budget and a compute budget, we give a non-adaptive privacy accounting procedure and practical heuristics to tune hyperparameters. This makes INVISIBLEINK usable off-the-shelf. In contrast, prior works [13, 19] require grid search to tune hyperparameters that affect the privacy-utility-compute tradeoff or estimates of the adaptive algorithm behavior (e.g. number of tokens generated) for practical deployment. In addition to increased computational cost, this can add to the privacy cost [20].

We also open source a pip-installable Python package (invink), available at https://github.com/cerai-iitm/invisibleink, to generate differentially private synthetic text using INVISIBLEINK. See §G for an example of the invink package in action.

---

[1]Generally, the drop in utility from DP can be offset by increasing data and computation, e.g., by averaging over larger batches to reduce sensitivity; DP model training also admits a similar phenomenon [15]. Thus, we compare methods by the computation cost necessary to attain given privacy and utility levels.

[2]We allow some baselines an advantage by (incorrectly) reducing their sensitivity by 2 or $\sqrt{2}$; see §5.

## 2 Related Work

Generating synthetic data with DP has an extensive literature, particularly over categorical features [21–25]. Many such approaches rely on the exponential mechanism [26–28]. Unfortunately, it is cryptographically hard to generate DP synthetic data that preserves all 2-way marginals with polynomial sample complexity [29]; see also [30, Thm. 6.12] and [31, Ch. 9]. Fortunately, public data can provide strong priors over data distributions to partially circumvent these lower bounds. Public data has long been used in private tabular data generation [32–35]. More recently, pretrained generative foundation models provide strong priors, enabling the DP generation of complex image/text data [36–41]. In this work, we focus on practical and scalable DP generation of text using LLMs.

We focus on practical and scalable DP text generation. DP text can be generated from DP fine-tuned models [42–46] or via private inference.[3] We focus on the latter as private fine-tuning can be prohibitively expensive, especially with large models [15]. Private inference is possible with various types of LLM access: via a text generation API [41, 46, 47], or white-box access to next-token logits [13, 47, 48]; we focus on the latter. Other DP text manipulation tasks include paraphrasing [49, 50] and next-token prediction [19, 51–54]. We adapt [19] to text generation as a baseline in §5.

Most prior approaches to DP text generation suffer from one or more restrictions: (1) a small number of generations, enough to serve as exemplars for in-context learning [48, 55, 56]; (2) short or highly constrained generations in classification or directed text generation tasks such as summarization or question-answering (as opposed to open-ended long-form generation) [13, 47]. Of these, only the API-access method of [41] and the white-box method of [13, 57] apply to long-form generation.

Prior DP text generation approaches diverge significantly from successful decoding algorithms in the NLP literature. Sampling from truncated next-token distributions [16, 58–62] is strongly backed by qualitative [16], quantitative [17, 18, 63], and theoretical [64] evidence; see the survey [8] for further details. INVISIBLEINK generates high-utility text by adapting truncated decoding to the DP setting.

The concurrent work of [57] extends the approach of Amin et al. [13] by clustering similar references into a batch. This orthogonal approach can also be integrated with INVISIBLEINK for synthetic text generation, though clustering may not be applicable to some tasks, such as RAG. [57] also replaces mean aggregation with median aggregation, with smaller local sensitivity and a tighter data-dependent *ex-post* DP guarantee [65] (evaluated after observing the algorithm's output). In contrast, INVISIBLEINK provides a stronger standard (data-independent and *ex-ante*) DP guarantee (evaluated before the algorithm is run).

## 3 Preliminaries: Differentially Private Text Generation

**Language Models & Decoding.** An auto-regressive language model $P$ over a vocabulary $V$ defines a distribution $P(x_t|\boldsymbol{x}_{<t})$ over the next token $x_t \in V$ in response to a prefix $\boldsymbol{x}_{<t} := (x_1, \ldots, x_{t-1}) \in V^*$; here, $V^*$ denotes the set of all sequences of elements from $V$. These probabilities are typically obtained as the softmax of the next-token logits $\phi(\cdot \,|\, \boldsymbol{x}_{<t})$ as $P(x_t|\boldsymbol{x}_{<t}) \propto \exp\left(\phi(x_t|\boldsymbol{x}_{<t})\right)$.

Given a query $\boldsymbol{q} \in V^*$ and a reference text $\boldsymbol{r} \in V^*$, we could iteratively sample a response $x_t \sim P(\cdot \,|\, \boldsymbol{q}, \boldsymbol{r}, \boldsymbol{x}_{<t})$ from the LLM $P$ with a concatenation of $\boldsymbol{q}, \boldsymbol{r}, \boldsymbol{x}_{<t}$ as context, and $\boldsymbol{x}_{<1} = \emptyset$. Unfortunately, this approach tends to over-represent incorrect low-probability tokens (as $|V| \sim O(10^5)$ or larger), leading to degenerate text [16, 17]. A common strategy is to use *decoding algorithms* that reshape $P(\cdot \,|\, \cdot)$ into another distribution $Q(\cdot \,|\, \cdot)$ from which we sample the next token.

Typical decoding algorithms promote more conservative outputs by suppressing the unreliable low-probability tokens. For example, *temperature rescaling* [66] at a temperature $\tau > 0$ rescales the logits, while *top-k sampling* [58] only samples from the $k$ largest logits. Mathematically,

$$Q_\tau(x_t|\boldsymbol{x}_{<t}) \propto \exp\left(\tfrac{1}{\tau}\phi(x_t|\boldsymbol{x}_{<t})\right), \text{ and } Q_{\text{top-}k}(x_t|\boldsymbol{x}_{<t}) \propto \begin{cases} P(x_t|\boldsymbol{x}_{<t}), & \text{if } x_t \in V_k, \\ 0, & \text{else}, \end{cases} \tag{1}$$

where $V_k \subset V$ is the set of $k$ tokens with highest logits $\phi(\cdot \,|\, \boldsymbol{x}_{<t})$. See Fig. 2 for an example.

---

[3]Private inference can be used on pretrained models prompted with private data (our setting) or non-DP fine-tuned models. Both are conceptually identical as they privatize a set of non-private predictions.

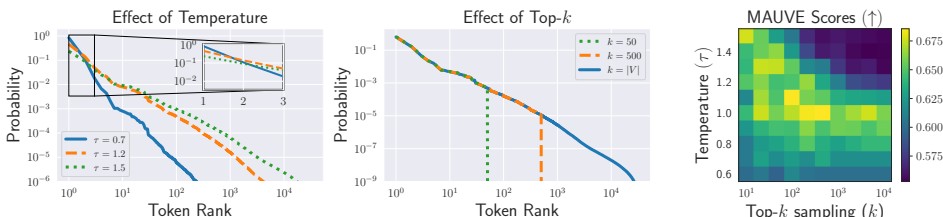

Figure 2: **Left & Center**: Illustration of how two common decoding algorithms—temperature rescaling and top-$k$ sampling—reshape the next-token probabilities for (non-private) LLM-based text generation. **Right**: Heatmap of MAUVE scores [17, 18] of synthetic text generated for the MIMIC-IV-Notes dataset (without using any sensitive references). The best generations (highest MAUVE scores) are obtained at $\tau \approx 1.1$ and $k \approx 100$; INVISIBLEINK exhibits similar behavior of decoding hyperparameters for private text generation.

The quality of non-DP long-form text generation is primarily determined by the decoding algorithm and its hyperparameter ($\tau$ or $k$) [16, 17, 63]. Typically, these hyperparameters are tuned to balance two competing objectives: reducing (or zeroing out) the probability of generating contextually inappropriate or low-probability tokens, while allowing the selection of moderately probable but contextually appropriate tokens to maintain diversity and coherence. In practice, this typically translates to a temperature $\tau \approx 1$ or slightly smaller, and $k \in [10, 100]$ (cf. Fig. 2).

In this work, we generate text with a set of privacy-sensitive references $\boldsymbol{R} = (\boldsymbol{r}_1, \dots, \boldsymbol{r}_B) \in (V^*)^B$. This setting arises in synthetic text generation, where we instruct an LLM to generate text similar to given references, or in RAG systems where $\boldsymbol{R}$ contains retrieved information over confidential knowledge bases. The model output may leak sensitive information from $\boldsymbol{R}$ in such cases.

**Differential Privacy (DP).** DP provides formal protections against such privacy leakage. At a high level, a (randomized) text generation algorithm $\mathcal{A}$ satisfies DP at the example level, if for all *adjacent* reference sets $\boldsymbol{R}, \boldsymbol{R}'$ (that differ in a single sensitive example), the probability distributions $\mathcal{A}(\boldsymbol{q}, \boldsymbol{R})$ and $\mathcal{A}(\boldsymbol{q}, \boldsymbol{R}')$ over responses $\boldsymbol{x} \in V^*$ are "nearly indistinguishable". A fully specified example-level DP guarantee requires precise notions of adjacency and indistinguishability. We focus on the **replace-by-null adjacency**, where an adjacent $\boldsymbol{R}'$ is obtained by replacing one element of $\boldsymbol{R}$ by the empty string (or vice versa); we denote it as $\boldsymbol{R} \simeq \boldsymbol{R}'$. This is functionally similar to adding or removing a reference, but simplifies the theoretical technicalities; see [15] or §B for a comparison of adjacency notions. We use the indistinguishability notion of zero-concentrated DP (zCDP) [67]. Let $D_\alpha$ denote the $\alpha$-Rényi divergence. Then, an algorithm $\mathcal{A}$ satisfies $\rho$-zCDP if

$$D_\alpha\big(\mathcal{A}(\boldsymbol{q}, \boldsymbol{R}) \| \mathcal{A}(\boldsymbol{q}, \boldsymbol{R}')\big) \leq \rho\,\alpha \quad \text{for all } \alpha > 1 \text{ and all adjacent } \boldsymbol{R} \simeq \boldsymbol{R}'.$$

Smaller values of $\rho$ mean that $\mathcal{A}(\boldsymbol{q}, \boldsymbol{R})$ and $\mathcal{A}(\boldsymbol{q}, \boldsymbol{R}')$ are closer to each other (in terms of the Rényi divergence) and the impact of the differing example between $\boldsymbol{R}, \boldsymbol{R}'$ on the output is low, indicating higher privacy. The zCDP guarantee can be converted into other notions, e.g., $(\varepsilon, \delta)$-DP [68–70]. Our techniques also translate directly to other notions of adjacency and indistinguishability; see §C.

**DP Text Generation via the Exponential Mechanism.** Our goal is to design an algorithm $\mathcal{A}(\boldsymbol{q}, \boldsymbol{R})$ to generate text in response to a given query $\boldsymbol{q}$ and references $\boldsymbol{R}$ with a desired $\rho$-zCDP guarantee. We assume the **whitebox setting** where model logits $\phi(\cdot \mid \boldsymbol{x})$ can be queried for any $\boldsymbol{x} \in V^*$. An increasing number of state-of-the-art general-purpose and reasoning open-weights models including DeepSeek, LLaMA3, Mistral, GPT-OSS allow whitebox access to model logits. This is in contrast to the more restrictive API-access setting, where one inference call returns an entire text sequence [41].

We describe the prior state-of-the-art approach of Amin et al. [13], which generates the next token $x_t \in V$ via the canonical exponential mechanism [14] over the next-token logits of the given LLM. Given a query $\boldsymbol{q}$ and sensitive references $\boldsymbol{R}$, the first step is to obtain the next-token logits $\boldsymbol{\phi}_i = \phi(\cdot \mid \boldsymbol{q}, \boldsymbol{r}_i, \boldsymbol{x}_{<t}) \in \mathbb{R}^{|V|}$ for each $i \in [B]$ with an LLM inference call. Second, we clip and aggregate the logits as $\bar{\boldsymbol{\phi}} = (1/B) \sum_{i=1}^B \mathsf{clip}_C(\boldsymbol{\phi}_i)$, where $\mathsf{clip}_C(\boldsymbol{\phi})$ clips each coordinate of $\boldsymbol{\phi}_i$ to lie in $[-C, C]$.[4] We can then sample a $\rho$-zCDP token $x_t \in V$ by sampling [71] from the distribution

$$Q(x_t \mid \boldsymbol{q}, \boldsymbol{R}, \boldsymbol{x}_{<t}) \propto \exp\left(\tfrac{\sqrt{2\rho}}{\mathsf{sens}(\boldsymbol{\phi})} \cdot \bar{\boldsymbol{\phi}}(x_t)\right) \tag{2}$$

---

[4] The clipping in [13] re-centers the logits (which are invariant to additive shifts). While this is empirically important, we omit the details in our exposition as the underlying mathematical properties do not change.

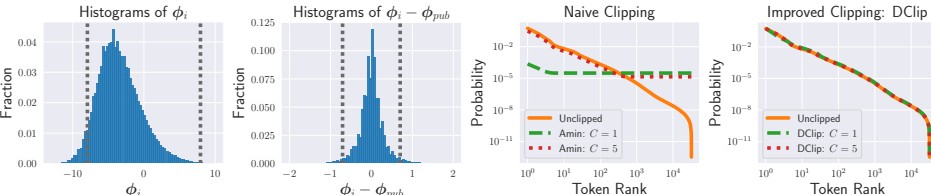

Figure 3: **Left two**: Histograms of private logits $\phi_i$ and differences from public logits $\phi_i - \phi_{\text{pub}}$ for a synthetic data sample generated from the MIMIC dataset, with $5^{\text{th}}$ and $95^{\text{th}}$ percentiles shown by the dotted lines. The spread of values for $\phi_i - \phi_{\text{pub}}$ is significantly smaller (around $10\times$) than that of $\phi_i$. Thus, $\text{DClip}_C(\phi_i, \phi_{\text{pub}})(y) = \phi_i(y)$ for over $95\%$ of all $y \in V$ with $C \approx 1$, while the naive clipping of Amin et al. [13] requires $C \approx 8$. This translates into an $8\times$ gain in computational efficiency. **Right two**: A small clip norm of $C = 1$ (using Amin et al. [13]'s method introduces significant bias resulting in a near-uniform distribution over the vocabulary. In contrast, DClip (see §4) preserves probabilities with minimal distortion even at $C = 1$.

where the sensitivity $\text{sens}(\bar{\phi})$ measures the maximum change in $\bar{\phi}(y)$, when we move to an adjacent set $\boldsymbol{R}' \simeq \boldsymbol{R}$ of references for any $y \in V$. The clipping and aggregating steps control the sensitivity as $\text{sens}(\bar{\phi}) = O(C/B)$,[5] so that Eq. (2) reduces to sampling with temperature $\tau = O(C/B\sqrt{\rho})$.

**Drawbacks of Prior Work.** If the clip norm $C$ is too small, the resulting distortions in the next-token logits boost the probabilities of the unlikely tail tokens (see Fig. 3), and if $C$ is too large, the temperature $\tau \propto C/B$ becomes too large. Both scenarios lead to degenerate text. Generating non-degenerate text from this method requires taking $C$ sufficiently large (e.g. Amin et al. [13] recommend $C \approx 10$), and $B$ large enough to ensure a temperature of $\tau = O(C/B\sqrt{\rho}) \approx 1$.

That is, generating each private token requires a large number $B$ of LLM next-token inference calls, which can be prohibitive if the minimum admissible $B$ is large. In practice, we can typically control $B$, as this is the number of reference documents we use for one synthetic generation, or the number of retrieved references in RAG systems. However, we find in §5 that the method of [13] fails to produce coherent text at a batch size $B$ smaller than 50 or 100, making it prohibitively expensive at scale.

Finally, Amin et al. [13] adaptively use tokens generated from public logits $\phi_{\text{pub}} = \phi(\cdot \,|\, \boldsymbol{q}, \boldsymbol{x}_{<t})$ using the sparse vector technique (SVT) [72] if they are close enough to the aggregated private logits $\bar{\phi}$. This improves the overall privacy-utility tradeoffs, but makes noise calibration challenging. The privacy guarantee scales with the number of *private tokens* adaptively generated (often unknown a priori), as opposed to the *total* number of generated tokens (a priori specified). This leaves two choices to calibrate the noise hyperparameters: (a) estimate the fraction of times SVT chooses private tokens, or (b) use grid search. Both options can incur increased computation and privacy costs [20].

Next, we describe how INVISIBLEINK overcomes these drawbacks.

## 4 INVISIBLEINK: DP Text Generation under Strict Compute Budgets

INVISIBLEINK aims to improve the privacy loss with a surgical clipping function, and the utility with approximating the top-$k$ decoding over *private* logits at any given computation budget.

**DClip: A Targeted Clipping Function.** The private next-token logits $\phi_i := \phi(\cdot \,|\, \boldsymbol{q}, \boldsymbol{r}_i, \boldsymbol{x}_{<t})$ encode not just the sensitive information in the reference $\boldsymbol{r}_i$, but also general information about language, grammar and syntax, semantics, a degree of world knowledge and common sense, and style/tone of the preceding tokens [e.g. 73–75]. Clipping the next-token logits involves paying a privacy cost for this general non-private information, leading to wasted privacy budget.

This non-private information is already captured by a pretrained model, even without access to the sensitive reference $\boldsymbol{r}_i$. We isolate and selectively clip only the extra information conveyed by the private logits $\phi_i$ over and above the public logits $\phi_{\text{pub}} := \phi(\cdot \,|\, \boldsymbol{q}, \boldsymbol{x}_{<t}) \in \mathbb{R}^{|V|}$, as follows:

$$\text{DClip}_C(\phi_i, \phi_{\text{pub}}) := \phi_{\text{pub}} + \text{clip}_C(\phi_i - \phi_{\text{pub}}), \tag{3}$$

---

[5] We have $\text{sens}(\bar{\phi}) = \frac{2C}{B}$ (resp. $\frac{C}{B}$) under replace-by-null (resp. zero-out) adjacency; cf. §B. We give this method an advantage in §5 by (incorrectly) taking $\text{sens}(\bar{\phi}) = C/B$ under replace-by-null adjacency.

---

**Algorithm 1** INVISIBLEINK for DP Text Generation

---

**Require:** LLM logit API $\phi(\cdot \mid \cdot)$, vocabulary $V$, query $\boldsymbol{q}$, sensitive references $\boldsymbol{R} = \{\boldsymbol{r}_1, \ldots, \boldsymbol{r}_B\}$, max text length $T$, clip norm $C$, temperature $\tau$, top-$k$ parameter, initial generation $\boldsymbol{x} = \emptyset$
1: **for** $t = 1, \ldots, T$ **do**
2:      For $i \in [B]$, set $\boldsymbol{\phi}_i = \phi(\cdot \mid \boldsymbol{q}, \boldsymbol{r}_i, \boldsymbol{x}_{<t}) \in \mathbb{R}^{|V|}$ and $\boldsymbol{\phi}_{\mathsf{pub}} = \phi(\cdot \mid \boldsymbol{q}, \boldsymbol{x}_{<t}) \in \mathbb{R}^{|V|}$      ▷ *LLM calls*
3:      $V_k^+ = \text{EXPANDEDTOPVOCABULARY}(\boldsymbol{\phi}_{\mathsf{pub}}, k, C, B)$      ▷ *Vocabulary for Top-$k$+ sampling*
4:      Set $\bar{\boldsymbol{\phi}} = \boldsymbol{\phi}_{\mathsf{pub}} + \frac{1}{B} \sum_{i=1}^{B} \mathsf{clip}_C(\boldsymbol{\phi}_i, \boldsymbol{\phi}_{\mathsf{pub}})$      ▷ *Aggregated clipped logits*
5:      Sample $x_t \sim \mathsf{softmax}(\bar{\boldsymbol{\phi}}[V_k^+]/\tau)$      ▷ *Exponential mechanism over $V_k^+$*
6:      **Yield** next token $x_t$, append it to $\boldsymbol{x}$, and break if $x_t = $ `<eos>`      ▷ *Generated Token*
7: **procedure** EXPANDEDTOPVOCABULARY$(\boldsymbol{\phi}_{\mathsf{pub}}, k, C, B)$
8:      Set $\ell$ to be the $k^{\text{th}}$ largest entry of $\boldsymbol{\phi}_{\mathsf{pub}}$      ▷ *top-$k$ threshold of $\boldsymbol{\phi}_{\mathsf{pub}}$*
9:      **Return** $\{y \in V : \boldsymbol{\phi}_{\mathsf{pub}}(y) \geq \ell - 2C/B\}$      ▷ *Expand top-$k$ threshold of $\boldsymbol{\phi}_{\mathsf{pub}}$ by $2C/B$*

---

where $C > 0$ is a specified clip norm, and $\mathsf{clip}_C(\boldsymbol{\phi})$ projects each coordinate of $\boldsymbol{\phi}$ onto the interval $[-C, C]$. The clipped and aggregated logits $\bar{\boldsymbol{\phi}}$ can then be a drop-in replacement in Eq. (2) is

$$\bar{\boldsymbol{\phi}} = \tfrac{1}{B} \sum_{i=1}^{B} \mathsf{DClip}_C(\boldsymbol{\phi}_i, \boldsymbol{\phi}_{\mathsf{pub}}) = \boldsymbol{\phi}_{\mathsf{pub}} + \tfrac{1}{B} \sum_{i=1}^{B} \mathsf{clip}_C(\boldsymbol{\phi}_i - \boldsymbol{\phi}_{\mathsf{pub}}). \tag{4}$$

Its sensitivity is also conveniently bounded as follows (see §C for a proof):

**Property 1.** *Fix a query $\boldsymbol{q}$ and prefix $\boldsymbol{x}_{<t}$, and denote $\boldsymbol{\phi}_i = \phi(\cdot \mid \boldsymbol{q}, \boldsymbol{r}_i, \boldsymbol{x}_{<t})$. Then, the sensitivity of the map $(\boldsymbol{r}_1, \ldots, \boldsymbol{r}_B) \mapsto \frac{1}{B} \sum_{i=1}^{B} \mathsf{DClip}_C(\boldsymbol{\phi}_i, \boldsymbol{\phi}_{\mathsf{pub}})$ under the replace-by-null adjacency is $C/B$.*

Fig. 3 shows that the spread of $\boldsymbol{\phi}_i - \boldsymbol{\phi}_{\mathsf{pub}}$ is typically much smaller than that of $\boldsymbol{\phi}_i$, since $\boldsymbol{\phi}_{\mathsf{pub}}$ already contains a strong prior on language. We utilize a smaller clip norm $C$ without distorting the model outputs, and hence a *smaller compute cost $B$* to maintain a temperature of $\tau \propto C/B$ close to 1.

**Truncated Top-$k$+ Sampling.** As discussed in §3, the degeneracies of sampling long-form text with temperature $\tau \lesssim 1$ can be fixed by truncated decoding [16]. We now adopt this to DP text generation. Define $V_k(\boldsymbol{\phi}) \subset V$ as the top-$k$ vocabulary corresponding to logits $\boldsymbol{\phi} \in \mathbb{R}^{|V|}$:

$$V_k(\boldsymbol{\phi}) = \{y_1, \ldots, y_k\} \quad \Longleftrightarrow \quad \boldsymbol{\phi}(y_1) \geq \cdots \geq \boldsymbol{\phi}(y_k) \geq \max_{y \in V \setminus V_k(\boldsymbol{\phi})} \boldsymbol{\phi}(y).$$

Vanilla (non-private) top-$k$ sampling from one logit vector $\boldsymbol{\phi}_i = \phi_i(\cdot \mid \boldsymbol{q}, \boldsymbol{r}_i, \boldsymbol{x}_{<t})$ would restrict the vocabulary to $V_k(\boldsymbol{\phi}_i)$. Our goal is to extend the truncation to union of the top-$k$ contributions of the $B$ logit vectors $\boldsymbol{\phi}_1, \ldots, \boldsymbol{\phi}_B$ in Eq. (4). To get the contribution of the logit vector $\boldsymbol{\phi}_i$ to Eq. (4), we plug in $\boldsymbol{\phi}_j = \boldsymbol{\phi}_{\mathsf{pub}}$ for $j \neq i$, since $\boldsymbol{\phi}_j$ does not contain any private information about $\boldsymbol{\phi}_i$, as follows:

$$\boldsymbol{\phi}_i^{\mathsf{clip}} := \boldsymbol{\phi}_{\mathsf{pub}} + \tfrac{1}{B} \mathsf{clip}_C(\boldsymbol{\phi}_i - \boldsymbol{\phi}_{\mathsf{pub}}).$$

Then, the top-$k$ vocabulary for private generation is simply the union $\bar{V}_k := \cup_{i=1}^{B} V_k(\boldsymbol{\phi}_i^{\mathsf{clip}})$.

Unfortunately, restricting our vocabulary to this privacy-sensitive top-$k$ set $\bar{V}_k$ fails to satisfy DP. Changing the dataset from $\boldsymbol{R}$ to an adjacent $\boldsymbol{R}'$, can cause a token $y \in V$ to "jump out" of the top-$k$ set $\bar{V}_k$. Its probability of being generated then goes from a non-zero value to a zero value, leading to a privacy loss of $\log(Q(y|\boldsymbol{q}, \boldsymbol{R}, \boldsymbol{x}_{<t})/Q(y|\boldsymbol{q}, \boldsymbol{R}', \boldsymbol{x}_{<t})) = \infty$ (see [e.g. 70, Def. 2] for definitions).

An alternative explored in previous work is to simply sample from the top-$k$ tokens from the public logits $\boldsymbol{\phi}_{\mathsf{pub}}$. This incurs no privacy cost as the restricted vocabulary is independent of the private logits but may fail to generate tokens that are rare in the pretraining data but common in the sensitive data $\boldsymbol{R}$. These are likely relevant to the data generation domain, and it is desirable to recover them.

To do so we build a tight superset of $\bar{V}_k = \cup_{i=1}^{B} V_k(\boldsymbol{\phi}_i^{\mathsf{clip}})$ based purely on the public logits $\boldsymbol{\phi}_{\mathsf{pub}}$. Observe that clipping ensures $|\boldsymbol{\phi}_i^{\mathsf{clip}}(y) - \boldsymbol{\phi}_{\mathsf{pub}}(y)| \leq C/B$ for each $i$. So, each $y \in V_k(\boldsymbol{\phi}_{\mathsf{pub}})$ satisfies $\boldsymbol{\phi}_i^{\mathsf{clip}}(y) + C/B \geq \ell$, where $\ell$ is the top-$k$ threshold of $\boldsymbol{\phi}_{\mathsf{pub}}$. Thus, the top-$k$ threshold $\ell_i$ of $\boldsymbol{\phi}_i^{\mathsf{clip}}$ is bounded as $\ell_i \geq \ell - C/B$. It thus follows that:

$$y \in V_k(\boldsymbol{\phi}_i^{\mathsf{clip}}) \iff \boldsymbol{\phi}_i^{\mathsf{clip}}(y) \geq \ell_i \implies \boldsymbol{\phi}_{\mathsf{pub}}(y) + \tfrac{C}{B} \geq \ell - \tfrac{C}{B}. \tag{5}$$

In other words, the private top-$k$ vocabulary $V_k(\boldsymbol{\phi}_i^{\mathsf{clip}})$ always lies in the set $V_k^+ := \{y \in V : \boldsymbol{\phi}_{\mathsf{pub}}(y) \geq \ell - 2C/B\}$. Since this is true for each $i \in [B]$, we have that $V_k^+$ is a superset of $\bar{V}_k = \cup_{i=1}^{B} V_k(\boldsymbol{\phi}_i^{\mathsf{clip}})$. We refer to sampling from the set $V_k^+$ as *Top-$k$+ sampling*.

Since $V_k^+$ is constructed without using the private logits, sampling from it does not incur any privacy cost. We find in §5 that $\left|V_k^+ - V_k(\phi_{\mathsf{pub}})\right| \approx 10 \ll |V|$, i.e. $V_k^+$ is indeed a *tight* superset of $\bar{V}_k$. We further develop the intuition on Top-$k+$ sampling using a toy example in §C.3.

**INVISIBLEINK: Text Generation & Privacy Accounting.** Our approach wraps the above two components in an exponential mechanism-based outer loop; see Algorithm 1. Crucially, Algorithm 1 does not involve any data-dependent privacy mechanisms. This lets us give a zCDP bound, building directly upon the zCDP analysis of the exponential mechanism [71] and adaptively composing [70] the per-token zCDP guarantee over $T$ tokens. Adaptive composition ensures that previously generated tokens can be reused safely in future steps without additional privacy cost; see §C for proof.

**Theorem 2.** *Algorithm 1 with a maximum token budget $T$, a clipping threshold $C$, a set $\boldsymbol{R}$ of $B = |\boldsymbol{R}|$ references, and temperature $\tau$ satisfies $\rho_{\mathsf{seq}}$-zCDP with $\rho_{\mathsf{seq}} = TC^2/(2B^2\tau^2)$.*

When running the algorithm in practice, we assume that we are given the privacy budget $\rho_{\mathsf{seq}}$, the maximum sequence length $T$, and the number of references $B$ (which fixes the compute budget). We recommend a default temperature of $\tau \approx 1$. Then, Theorem 2 lets us set the clip norm $C = B\tau\sqrt{2\rho_{\mathsf{seq}}/T}$ to get the desired privacy guarantee (independent of the top-$k$ parameter).

Another advantage of our method is that generation gracefully falls back on that of the public model if the compute or privacy budget is too small. In this case, we get a very small $C$ and $\bar{\phi}_i \approx \phi_{\mathsf{pub}}$. This nice property is not satisfied by existing baselines. For example, as the privacy or computation budget gets smaller, [13] requires a small clip norm $C$, reducing their method to uniform sampling over the full vocabulary (see Fig. 3). This leads to gibberish text.[6] Similarly, AdaPMixED [19] has a data-dependent privacy guarantee that depends on the number of tokens sampled per generation. As the privacy or computation budget gets smaller, the length of synthetic generations falls sharply.

Finally, we note that it might be possible to improve the dependence of Theorem 2 on the token budget $T$ with regularization [e.g. 76, 77]; we leave this for future work.

## 5 Experiments

**Setup.** Given a reference dataset $\mathcal{D} = \{\boldsymbol{r}_1, \ldots, \boldsymbol{r}_N\}$, our task is to create $n$ synthetic texts $\boldsymbol{x}_1, \ldots, \boldsymbol{x}_n$ that are statistically similar to $\mathcal{D}$ while satisfying a given $\rho_{\mathsf{seq}}$-zCDP guarantee with respect to $\mathcal{D}$. We generate synthetic example $\boldsymbol{x}_j$ based on a batch $\boldsymbol{R}_j = (\boldsymbol{r}_{jB+1}, \ldots, \boldsymbol{r}_{(j+1)B})$ of sensitive references with a query $\boldsymbol{q}$ instructing a generation similar to the given references. By parallel composition, each example $\boldsymbol{x}_j$ must also satisfy $\rho_{\mathsf{seq}}$-zCDP. Since the batch size $B$ fixes the computation budget, we allow $(B+1)$ LLM next-token inference calls per generated token. This also limits us to $n \leq \lfloor N/B \rfloor$ synthetic examples. See §D for additional details on the task and experimental setup.

We report a central DP guarantee for a single hyperparameter setting to generate $n$ synthetic text samples; the randomized mechanism here is a set of $n$ autoregressive text generations from LLMs, which output a sequence of tokens[7] each. We propose heuristics for choosing optimal hyperparameters in §5.3 in lieu of accounting for privacy lost in hyperparameter tuning. Throughout this work, we report example-level (an example is a single text sample drawn from the reference dataset) $\rho$-zCDP guarantees (converted to $(\varepsilon, \delta)$-DP where necessary) under the *replace-by-null* adjacency; see §B for details. We release our code at: https://github.com/cerai-iitm/InvisibleInk-Experiments.

**Datasets.** We experiment with three datasets in clinical, legal, and commercial domains, see Tab. 1. MIMIC-IV-Note [78, 79] is a de-identified collection of medical text associated with ICU patients; we use the discharge summaries, which contain sensitive diagnosis, treatment, and medication information. Further, this dataset is legally unavailable for LLM pretraining, making it valuable in modeling real-world distribution-shift sensitive domains. The Text Anonymization Benchmark (TAB) [80] contains $N = 1013$ (training set) court case notes (in English) from the European Court of Human Rights (ECHR) with personal identifiers and other confidential attributes. This low-resource dataset only admits extremely small batch sizes $B$ (e.g., $B = 127$ gives us a maximum of 8 synthetic texts). The Yelp Reviews dataset [81] contains user-generated reviews and ratings for businesses with

---

[6]For example, the word "differential" might be made up of two tokens "differ" and "-ential". Sampling uniformly at random could yield the "-ential" token without an appropriate preceding token, producing gibberish.

[7]The token string ends in an `<EOS>` token unless the number of tokens generated reaches the limit $T$ first.

| Dataset | Domain | Text Type | Size | Avg. Length | # Generations | Max. Length |
|---------|--------|-----------|------|-------------|---------------|-------------|
| MIMIC-IV-Note | Medical | Discharge Summaries | 311K | 298.4 | 1000 | 500 |
| Yelp | Commercial | Reviews | 1.9M | 145.2 | 500 | 200 |
| TAB | Legal | Court Case Texts | 1013 | 387.7 | 100 | 500 |

Table 1: Dataset summaries. Lengths are measured in tokens and we use at most $Bn \leq 128K$ reference texts.

personal opinions and location references, and is a standard benchmark in DP text generation [13, 41]. We intentionally choose real privacy-sensitive datasets (e.g. MIMIC, TAB) datasets with long-form text, rather than standard internet datasets (AGNews, IMDB, etc.) that are well-represented in the pretraining data of most LLMs [82, 83]. See §D.2 for further discussion.

**Models.** We primarily use the TinyLLaMA 1.1B [84], a compact open-weights model with strong performance in compute-constrained settings. We also use LLaMA3.2-1B [85] to study the effect of large vocabularies ($|V| = 128K$ vs. $32K$ for TinyLlama), and LLaMA3-8B [86] for scaling.

**Baselines and Hyperparameters.** We compare INVISIBLEINK (with and without Top-$k$+ sampling) with the prior white-box SoTA methods Amin et al. [13] for DP text generation and AdaPMixED [19] for DP next-token prediction. We iteratively sample from the predicted next-token distribution of AdaPMixED to adapt it to DP text generation. For all methods, we fix the batch size $B$ as it fixes the computation budget. We select other hyperparameters, such as the clip norm $C$ for INVISIBLEINK and Amin et al. [13] to achieve a desired privacy guarantee. See §D.5 for further details.

The method of Amin et al. [13] has a sensitivity of $C/B$ under the zero-out adjacency, compared to $2C/B$ under the *replace-by-null* adjacency (see Theorem 9 of §B). Yet, we give it an advantage by (incorrectly) taking the sensitivity to be $C/B$ under the *replace-by-null* adjacency. AdaPMixED originally gives a data-dependent DP guarantee under the add-or-remove adjacency in [19]. Since we find that the utility trends under both adjacency notions are qualitatively similar (see §E.6), we modify AdaPMixED directly and report the results for *replace-by-null* adjacency here.

The DP guarantees of [13] are calibrated on the number of times the adaptive SVT step (whose output is data-dependent) selects the private tokens. Further, AdaPMixED [19] gives data-dependent DP guarantees. This makes precise *pre-hoc* privacy accounting difficult. In our experiments we select hyperparameters that give the desired DP guarantee "on average". For example, we find that the adaptive SVT step in [13] selects private tokens around $25\%$ of the time, so we calibrate their method to produce $\approx T/4$ private tokens and generate until the privacy budget is exhausted. We find that the privacy loss always exhausts the maximum budget for at least one batch across all settings.

We also compare INVISIBLEINK with the API-access method of Xie et al. [41] (AugPE). This restrictive setting naturally makes privatization harder. AugPE makes two types of LLM inference calls: generation and paraphrasing. We cannot make an apples-to-apples comparison between the computational cost of INVISIBLEINK to AugPE using the number of LLM inferences per generated tokens since AugPE is calibrated using the number of paraphrasing iterations made per generated sequence. Instead, we directly use the wall-clock runtime of both methods as a measure of computation cost. We also allow AugPE an advantage of $\sqrt{2}$ in the sensitivity under *replace-by-null* adjacency (see §B.3.3). For additional results and detailed generation settings, see §E.1.

**Metrics.** We evaluate the statistical similarity between the generated text and reference distribution with MAUVE scores [17, 18, 87], the gap between the average perplexities of generated and reference texts (denoted $\Delta$PPL), and average generation lengths to ensure that sufficient tokens are generated before exhausting the privacy budgets. The MIMIC discharge notes also contain medical named-entities such as names of medical conditions, procedures, and medicines. We plot the number of such entities identified by a MedNER model [88], as a measure of the domain-relevance of generated text.

## 5.1 Experimental Results

We compare the privacy-utility-compute tradeoffs of INVISIBLEINK with the baselines at a target level of $(\varepsilon = 10, \delta = 10^{-6})$-DP in Fig. 4. See Tab. 2 for evaluations at other $\varepsilon$ values (with $\delta = 10^{-6}$).

We find that INVISIBLEINK with Top-$k$+ sampling consistently produces the *highest quality text* with the longest generations (and the most medical named entities for MIMIC) at *smaller compute budgets*

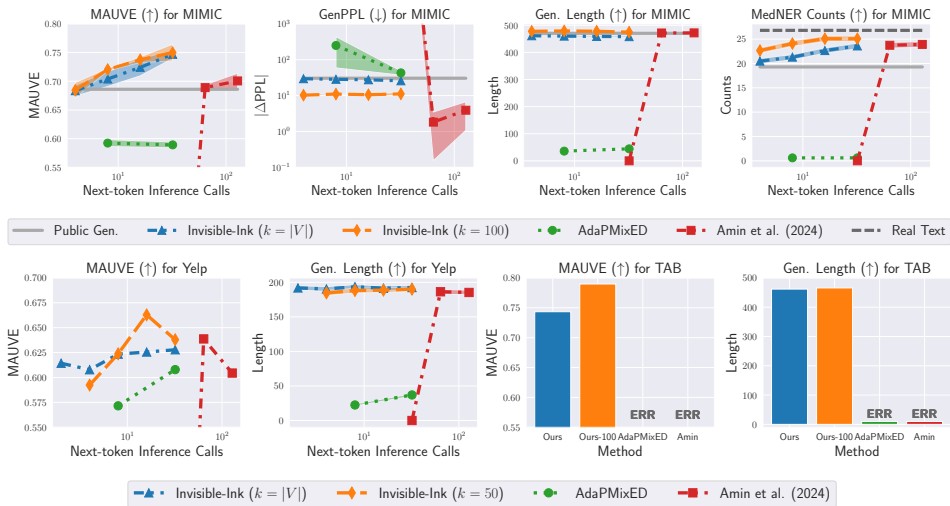

Figure 4: Utility-compute tradeoffs at $(\varepsilon = 10, \delta = 10^{-6})$ DP on each dataset across varying compute budget from $B \in \{1, 3, \ldots, 127\}$. Results reported over 3 runs with $95\%$ confidence intervals (see §D.6) for MIMIC and 1 run for Yelp/TAB datasets. INVISIBLEINK produces text that matches or exceeds the baselines at a fraction of the compute. The baselines do not even work for the low-resource TAB dataset at small batch sizes $B = 7$.

| | INVISIBLEINK ($k = 100$) | | | | INVISIBLEINK ($k = |V|$) | | | Amin et al. [13] | | | AdaPMixED [19] | | |
|---|---|---|---|---|---|---|---|---|---|---|---|---|---|
| $B$ | 3 | 7 | 15 | 31 | 3 | 7 | 15 | 31 | 63 | 127 | 7 | 31 | 127 |
| $\varepsilon = 1$ | 68.3 $_{0.1}$ | 66.9 $_{1.1}$ | 67.9 $_{1.5}$ | **69.2** $_{1.3}$ | 67.3 $_{0.9}$ | 67.6 $_{0.9}$ | 68.2 $_{1.4}$ | INF | INF | INF | 58.8 $_{0.3}$ | 56.8 $_{0.1}$ | 59.2 $_{0.2}$ |
| $\varepsilon = 3$ | 67.4 $_{1.1}$ | 69.0 $_{0.7}$ | 69.7 $_{0.6}$ | **70.9** $_{1.2}$ | 66.9 $_{2.0}$ | 69.1 $_{1.3}$ | 69.1 $_{0.7}$ | INF | INF | 55.2 $_{0.0}$ | 58.8 $_{0.1}$ | 58.9 $_{0.3}$ | 64.7 $_{1.0}$ |
| $\varepsilon = 5$ | 67.7 $_{0.1}$ | 70.3 $_{1.0}$ | 69.7 $_{0.9}$ | **73.5** $_{1.0}$ | 68.6 $_{0.5}$ | 68.9 $_{0.7}$ | 69.0 $_{0.7}$ | INF | INF | 69.5 $_{0.0}$ | 59.4 $_{0.3}$ | 59.8 $_{0.4}$ | 69.9 $_{0.9}$ |
| $\varepsilon = 10$ | 68.4 $_{1.4}$ | 72.0 $_{0.2}$ | 73.7 $_{0.4}$ | **75.0** $_{1.5}$ | 68.3 $_{0.9}$ | 70.4 $_{1.3}$ | 72.4 $_{1.4}$ | INF | 68.9 $_{0.5}$ | 70.1 $_{1.3}$ | 59.2 $_{0.5}$ | 58.9 $_{0.5}$ | TLE |

Table 2: MAUVE (%) scores, reported with mean and $95\%$ confidence intervals (see §D.6) over 3 runs, for 1000 synthetic generations of the MIMIC dataset using TinyLLaMA. Synthetic generation without any private references ($\varepsilon = 0$) yields a MAUVE of **68.56%** (best hyperparameters). INVISIBLEINK outperforms every private baseline with $\approx 8\times$ **smaller batch size**. The results for INVISIBLEINK with $k = |V|$ are reported for $\tau = 1.0$ and for $k = 100$ are reported for $\tau = 1.1$. AdaPMixED [19] sometimes exceeds a wall clock time of 36 hours (denoted **TLE**- Time Limit Exceeded); Amin et al. [13]'s method fails to generate any synthetic text for small privacy/compute budgets (denoted **INF**- infeasible). The top two scores for every $\varepsilon$ are highlighted.

across datasets. Consider the results on MIMIC, for example: INVISIBLEINK can achieve a MAUVE score of $\approx 71\text{-}74\%$ with 8-16 inference calls per token, while Amin et al. [13] achieves $\approx 70\text{-}72\%$ at with 64-128 inference calls: this is an $8\times$ **improvement in computation cost**. INVISIBLEINK also produces more medical named entities than Amin et al. [13] capturing the domain-specific information in the sensitive references and producing qualitatively better text.

Tab. 2 shows that INVISIBLEINK exhibits graceful degradation in utility when operating under tight privacy budgets or compute constraints, where baselines fail. Specifically, Amin et al. [13]'s method uses up all its privacy budget for the SVT term and cannot produce any text at (a) $B \leq 31$, (b) $\varepsilon \leq 5$ and $B \leq 63$, and (c) $\varepsilon = 1$ and $B \leq 127$; see §E. Meanwhile, AdaPMixED extinguishes its data-dependent privacy budget after generating as few as 10 tokens for $B = 8$. Both baselines require large batch sizes to operate meaningfully, rendering them impractical in compute-constrained environments. We discuss the full sets of hyperparameters explored, in §D.5.

**Comparison with API-Access Methods.** We find that INVISIBLEINK completely outperforms the API-access based method AugPE [41] across privacy budget settings and evaluation metrics in Fig. 5. We remark that this is expected due to the more restrictive API-access setting of AugPE. Interestingly, we found in preliminary experiments that AugPE showed competitive (with INVISIBLEINK) performance on the Yelp dataset. This can likely be attributed to the LLM-based paraphrasing step working best when the dataset of private references is well-represented in the pre-training data. See §E.1, for more details and additional results.

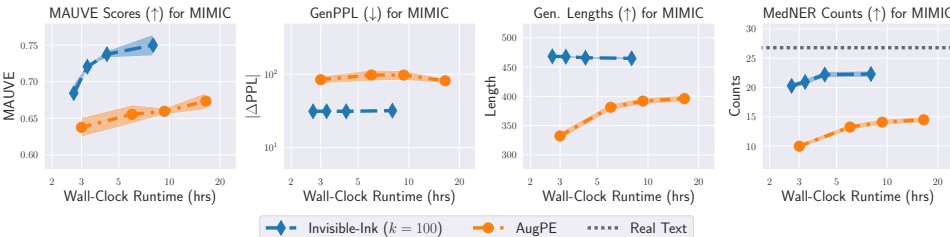

Figure 5: **INVISIBLEINK outperforms the API-access method of AugPE [41] across all settings:** Utility vs Compute plots (avg. over 3 runs with 95% confidence intervals) for INVISIBLEINK and AugPE for $\varepsilon = 10$ for 1000 synthetic texts generated for the MIMIC dataset. Wall-clock run time is used as a proxy for computational cost. We report results for $B + 1 = 4, 8, 16, 32$ for INVISIBLEINK and and $T_{\text{AugPE}} = 1, 3, 5, 10$ for AugPE.

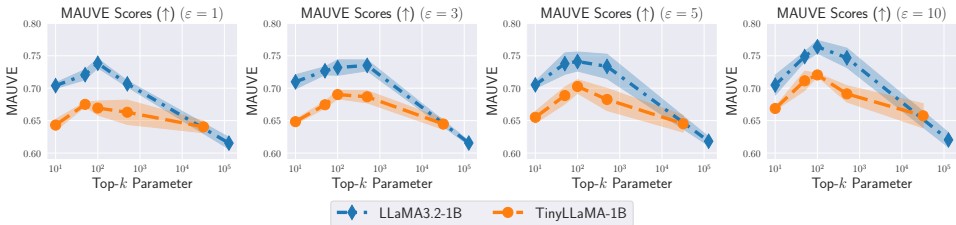

Figure 6: **Truncated decoding offers significant benefits**: MAUVE scores (avg. over 3 runs with 95% confidence intervals) for DP synthetic text generation of 1000 samples from the MIMIC Dataset using TinyLLaMA-1B and LLaMA3.2-1B models at a temperature $\tau = 1.1$ and batch size $B = 7$.

## 5.2 Understanding INVISIBLEINK: Ablations, Scaling and Additional Experiments

We examine the importance of the two key components—DClip and the Top-$k+$ sampling—of INVISIBLEINK. Fig. 3 shows that the smaller active range of $\phi_i - \phi_{\text{pub}}$ allows us to use a smaller $C$, when clipping using DClip, to maintain similar levels of distortion. This translates into an $\approx 8\times$ improvement (see Fig. 4) in computation cost at the same privacy-utility levels. Fig. 6 shows us that Top-$k+$ decoding offers significant improvements (in MAUVE scores) over using the full vocabulary (72% vs. 65% at $\varepsilon = 10$ for TinyLlama). This distinction is more pronounced for the similar-sized Llama-3.2 1B model (76% vs. 62% at $\varepsilon = 10$), with a $4\times$ larger vocabulary ($128K$ vs. $32K$).

**Additional Experiments.** We present additional statistics of Top-$k+$ sampling from the expansion set $(V_k^+ \setminus V_k)$, in §E.3 and show empirically that $V_k^+$ is a tight superset of $V_k$. We also conduct additional experiments evaluating the suitability of private synthetic text for a downstream classification task on the Yelp dataset in §E.2. To highlight the scalability of INVISIBLEINK to larger models we generate synthetic text using a LLaMA3 8B model at low batch sizes (incompatible with baselines) in §E.4. §E.5 gives additional ablations on the effect of the temperature, clip norm, and top-$k$ parameters. In particular, we show that setting the temperature $\tau \in [1.0, 1.1]$ and the top-$k$ parameter $k \approx 100$ yields near-optimal generations across settings. As noted previously, a comparison of the performance of AdaPMixED for *replace-by-null* and *add-or-remove* adjacency is also presented in §E.6 to show that the quantitative results for either notion follows similar trends.

## 5.3 Practical Recommendations

We recommend tuning a temperature parameter $\tau \approx 1$ (generally in the range of 0.9 to 1.2), if permitted by the computation budget. A reasonable heuristic to avoid tuning is to directly set $\tau = 1$, as it achieves competitive performance. Then, we set a clip norm $C = B\tau / \sqrt{2\rho_{\text{seq}}/T}$, where the batch size $B$ fixes the computation budget, $\rho_{\text{seq}}$-zCDP is the target privacy guarantee, and $T$ is the maximum token budget. Finally, we must tune a free parameter $k$ for Top-$k+$ sampling. We recommend tuning $k$ following best practices for non-private decoding (e.g., $k \in [50, 100]$ for open-ended settings and smaller for more directed settings such as reasoning).

**Conclusion.** We introduce INVISIBLEINK, a scalable framework for long-form text generation that provides rigorous differential privacy guarantees with respect to sensitive references. For a detailed discussion of the limitations and broader impact of our work, see §F.

## Acknowledgments

The authors thank Ambreesh Parthasarathy, Krithi Shailya, Danish Pruthi, and Balaraman Ravindran for fruitful discussions, as well as the authors of [13] for constructive discussions. VV is supported by the Post-Baccalaureate Fellowship at the Centre for Responsible AI (CeRAI), IIT Madras. KP is grateful for funding from ANRF's PM ECRG (ANRF/ECRG/2024/004321/PMS), Schmidt Sciences' AI2050 program, the startup compute grant of the Wadhwani School of Data Science & AI (WSAI), IIT Madras, and faculty research awards.

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

# Appendix

## Table of Contents

# A   Notation

We summarize our main notation in Tab. 3.

| Symbol | Description |
|---|---|
| $V$ | LLM vocabulary, set of all allowed tokens |
| $V^*$ | set of all sequences of tokens (of any length) |
| $\boldsymbol{x}_{<t}$ | $= (x_1, \ldots, x_{t-1}) \in V^{t-1}$, all previously generated tokens before time-step $t$-token |
| $\phi$ | LLM logit function, where $\phi(x_t \mid \boldsymbol{x}_{<t}) \in \mathbb{R}$ denotes the logit score of the next token $x_t \in V$ given the prefix $\boldsymbol{x}_{<t}$ |
| $\phi(\cdot \mid \boldsymbol{x}_{<t})$ | the vector of next-token logits in $\mathbb{R}^{|V|}$ indexed by the token $y \in V$ |
| $\boldsymbol{q}$ | query or prompt for generation (not privacy-sensitive) |
| $\boldsymbol{r}$ | reference text (privacy sensitive) |
| $\boldsymbol{R}$ | batch of sensitive references $\boldsymbol{R} = \{\boldsymbol{r}_1, \ldots, \boldsymbol{r}_B\}$ |
| $\mathcal{D}$ | dataset of sensitive references |
| $N$ | $= |\mathcal{D}|$, dataset size |
| $n$ | number of generations |
| $\boldsymbol{\phi}_{\mathsf{pub}}$ | Logit vector $\phi(\cdot \mid \boldsymbol{q}, \boldsymbol{x}_{<t}) \in \mathbb{R}^{|V|}$ generated without using any sensitive references from $\boldsymbol{R}$ |
| $\boldsymbol{\phi}_i$ | Logit vector $\phi(\cdot \mid \boldsymbol{q}, \boldsymbol{r}_i, \boldsymbol{x}_{<t}) \in \mathbb{R}^{|V|}$ generated using sensitive reference $\boldsymbol{r}_i$ |
| $\mathsf{clip}_C(\boldsymbol{\phi})$ | clips its input vector $\boldsymbol{\phi}$ component-wise to have $\ell_\infty$ norm at most $C$ |
| $\mathsf{DClip}_C(\boldsymbol{\phi}, \boldsymbol{\phi}_{\mathsf{pub}})$ | $= \boldsymbol{\phi}_{\mathsf{pub}} + \mathsf{clip}_C(\boldsymbol{\phi} - \boldsymbol{\phi}_{\mathsf{pub}})$, our proposed clipping function that clips the difference with public logits |
| $V_k$ | set of $k$ tokens with highest public logits $\boldsymbol{\phi}_{\mathsf{pub}}$ |
| $V_k^+$ | expanded top-$k$ set of tokens; see §4 |
| $B$ | $= |\boldsymbol{R}|$, batch size; controls the computational cost of INVISIBLEINK |
| $C$ | clipping norm |
| $\tau$ | sampling temperature |
| $T$ | maximum number of tokens allowed per generation |

Table 3: A summary of all important notation used throughout the paper.

# B   Adjacency Notions and DP Guarantees for LLM Inference

The goal of this section is to clarify the technicalities of the provided DP guarantees for LLM inference and private text generation. In particular, we describe the adjacency notions used (implicitly or explicitly) in prior work and how they differ from the ones used in this work. We also describe how the advantage we give to the baselines in our experimental comparisons of §5.

A DP algorithm must satisfy the guarantee that changing the input dataset by one unit of data leads to a nearly indistinguishable output. To fully specify this guarantee, we need to mention each of the underlined quantities [15].

**Adjacency notion.** It describes what it means to change the input dataset (e.g., add/remove/replace some data). We explain this in detail in §B.1.

**Privacy unit.** The unit of data that is protected under the DP definition is known as the privacy unit. This quantity determines whether we protect individual examples (or documents; known as example-level DP), or all examples contributed by a single "user" (known as user-level DP). In this work, we only consider example-level DP.

**Indistinguishability Notion.** This refers to the specific way we quantify how similar the output distributions on two adjacent datasets are. We use zero-Concentrated DP (zCDP) throughout:

**Definition 3** (Zero-concentrated DP). *An algorithm $\mathcal{A}$ satisfies $\rho$-zCDP if*

$$D_\alpha\big(\mathcal{A}(\boldsymbol{q}, \boldsymbol{R})\|\mathcal{A}(\boldsymbol{q}, \boldsymbol{R}')\big) \leq \rho\, \alpha \quad \text{for all } \alpha > 1 \text{ and all adjacent } \boldsymbol{R}, \boldsymbol{R}', \tag{6}$$

*where $D_\alpha$ denotes the $\alpha$-Rényi divergence between probability distributions*

$$D_\alpha(P\|Q) = \frac{1}{\alpha - 1} \log\big(\mathbb{E}_{X \sim Q}\left[\left(P(X)/Q(X)\right)^\alpha\right]\big)\,.$$

Conversion formulae between various notions of indistinguishability are known [e.g. 89, Fig. 2]. For example, a $\rho$-zCDP guarantee implies $(\varepsilon, \delta)$-DP with [69, Thm. 21]

$$\varepsilon \leq \inf_{\alpha > 1} \left\{ \alpha\rho + \frac{1}{\alpha - 1} \log\frac{1}{\alpha\delta} + \log(1 - \alpha^{-1}) \right\} \leq \rho + 2\sqrt{\rho \log(1/\delta)}\,.$$

## B.1 Adjacency Notions in LLM Inference

We explain the different adjacency notions that might arise when the DP definition is instantiated in the context of LLM inference. Recall that we wish to generate text in response to a query $\boldsymbol{q}$ using sensitive references $\boldsymbol{R} \subset V^*$.

We define various notions of adjacency between two sets of sensitive references $\boldsymbol{R}, \boldsymbol{R}'$. Arguably, the most commonly used one in the literature is the add-remove adjacency, where the adjacent dataset is obtained by adding or removing one datapoint.

**Definition 4** (Add-or-Remove Adjacency). *Two sets $\boldsymbol{R}, \boldsymbol{R}' \subset V^*$ are said to be add-or-remove adjacent if $\boldsymbol{R}' = \boldsymbol{R} \cup \{\boldsymbol{r}\}$ or $\boldsymbol{R} = \boldsymbol{R}' \cup \{\boldsymbol{r}\}$ for some $\boldsymbol{r} \in V^*$.*

This notion of adjacency is semantically meaningful and maps to intuitive expectations of a privacy guarantee. Unfortunately, it can be technically challenging to develop rigorous algorithms in this setting. In particular, we cannot assume the sensitive dataset size $|\boldsymbol{R}|$ to be fixed, as adjacent $\boldsymbol{R}, \boldsymbol{R}'$ are of different sizes. This makes the dataset size a privacy-sensitive quantity that needs to be protected with privacy mechanisms; see [90, §E], [15, §2.1.1], and [91] for further discussions.

For example, the sensitivity of the mean operation $(s_1, \ldots, s_n) \mapsto (1/n) \sum_{i=1}^n s_i$ is a function of the dataset size $n$, meaning that we cannot directly use it in the add-or-remove paradigm. A common workaround (used, for example, in private learning with DP-SGD) is to privatize the sum query $(s_1, \ldots, s_n) \mapsto \sum_{i=1}^n s_i$ and post-process it to obtain an estimator of the mean. Unfortunately, this is not possible in the setting of §3: the logit vector $\bar{\phi}$ in Eq. (2) is actually obtained by averaging, and we cannot directly use the above trick here.

A common workaround to avoid these technical challenges with the add-or-remove adjacency is to use the so-called replace-by-null adjacency, where the removal of an element is simulated by swapping it with a special (problem-dependent) null element [e.g. 70, Sec. 6.2]. We instantiate the replace-by-null adjacency in the context of LLM inference in this work by choosing the null element as the empty string:

**Definition 5** (Replace-by-Null Adjacency). *Two sets $\boldsymbol{R} = \{\boldsymbol{r}_1, \ldots, \boldsymbol{r}_n\}$ and $\boldsymbol{R}' = \{\boldsymbol{r}'_1, \ldots, \boldsymbol{r}'_n\}$ of equal size $n$ are said to be adjacent in the replace-by-null model of adjacency if there exists an index $j \in [n]$ such that $\boldsymbol{r}_i = \boldsymbol{r}'_i$ for all $i \neq j$ and one of $\boldsymbol{r}_j, \boldsymbol{r}'_j$ equal the empty string $\emptyset$.*

Another common way to instantiate the null element leads to the so-called zero-out adjacency:

**Definition 6** (Zero-Out Adjacency). *Let $\perp$ denote a special null element such that the logits $\phi(y \mid \boldsymbol{q}, \boldsymbol{r} = \perp, \boldsymbol{x}_{<t}) = 0$ are always equal to zero for any token $y \in V$, query $\boldsymbol{q} \in V^*$, and prefix $\boldsymbol{x}_{<t}$.[8] Two sets $\boldsymbol{R} = \{\boldsymbol{r}_1, \ldots, \boldsymbol{r}_n\}$ and $\boldsymbol{R}' = \{\boldsymbol{r}'_1, \ldots, \boldsymbol{r}'_n\}$ of equal size $n$ are said to be adjacent in the zero-out notion of adjacency if there exists an index $j \in [n]$ such that $\boldsymbol{r}_i = \boldsymbol{r}'_i$ for all $i \neq j$ and one of $\boldsymbol{r}_j, \boldsymbol{r}'_j$ equal this null element $\perp$.*

---

[8]To be fully formal, the space of all possible reference sequences $\boldsymbol{r}$ should be enlarged to $V^* \cup \{\perp\}$ to accommodate the null element $\perp$.

Note that a zero logit vector implies next-token probabilities that are uniform over the vocabulary. As argued in §4, this is rather unrealistic as it could produce incomplete word pieces, while $r = \emptyset$ as the empty string still gives reasonable next-token predictions using the pretrained model's language and world knowledge. Further, the empty string of Theorem 5 is a semantically more meaningful null element than an artificially created $\perp$ element from Theorem 6. Thus, we primarily adopt the replace-by-null adjacency in this work.

**Remark 7.** *The replace-one (or swap) notion of adjacency is also commonly studied. Here, one element of $\boldsymbol{R}$ is replaced by any other arbitrary element to get $\boldsymbol{R}'$. It provides an $\approx 2\times$ stronger privacy guarantee at the same $\varepsilon$ (usually at the cost of a larger noise multiplier); see [15] for a discussion.*

### B.2 Sensitivity for the Exponential Mechanism

Throughout this work, we consider the following notion of sensitivity in the context of the exponential mechanism. Note that the sensitivity depends on the exact notion of adjacency considered (§B.1).

**Definition 8** (Sensitivity). *The $\ell_\infty$-sensitivity, referred throughout as simply the sensitivity, of a function $f : S \to \mathbb{R}^d$ (for any set $S$ and any output dimension $d$) under an adjacency relation "$\simeq$" is defined as:*

$$\mathsf{sens}(f) := \max_{\boldsymbol{R} \simeq \boldsymbol{R}'} \|f(\boldsymbol{R}) - f(\boldsymbol{R}')\|_\infty = \max_{\boldsymbol{R} \simeq \boldsymbol{R}'} \max_{i=1,\dots,d} \left| [f(\boldsymbol{R})]_i - [f(\boldsymbol{R}')]_i \right|.$$

The exponential mechanism with a target privacy level of $\rho$-zCDP selects an item $i \in \{1, \dots, d\}$ with respective scores $[f(\boldsymbol{R})]_i$ with probability [71]

$$\frac{\exp\left(\frac{\sqrt{2\rho}}{\mathsf{sens}(f)}[f(\boldsymbol{R})]_i\right)}{\sum_{j=1}^d \exp\left(\frac{\sqrt{2\rho}}{\mathsf{sens}(f)}[f(\boldsymbol{R})]_j\right)}. \tag{7}$$

Recall that we wish to generate text in response to a query $\boldsymbol{q}$ using sensitive references $\boldsymbol{R} \subset V^*$. In this context, Theorem 8 is instantiated with $f(\boldsymbol{R}) = \phi(\cdot \mid \boldsymbol{q}, \boldsymbol{R}, \boldsymbol{x}_{<t}) \in \mathbb{R}^{|V|}$, where we index an element in $\mathbb{R}^{|V|}$ with a corresponding token $y \in V$.

### B.3 Technical Details of DP in Baselines

In this subsection, we clarify the notions of adjacency used in prior works.

#### B.3.1 Adjacency Notion of Amin et al. [13] and Theory-Implementation Gap

The theoretical analysis of Amin et al. [13] (in particular, their Theorem 1) operates under the add-or-remove notion of adjacency (Theorem 4). As discussed in §B.1, aggregating the clipped logits using the mean operation requires particular care as the size of adjacent datasets can be different. Amin et al. [13] address this issue with two changes:

(a) they use batches of variable sizes (e.g. via hashing datapoints to batches); and
(b) they use the re-scaled sum query $(s_1, \dots, s_B) \mapsto (1/G) \sum_{i=1}^B s_i$, where $G$ is a guess of the (variable and possibly random) batch size $B$.

To ensure correctness of the privacy guarantee, the scaling factor $G$ must be fixed a priori and in a data-independent manner. The accuracy of the guess $G$ (relative to the true batch size $B$) does not impact the correctness of the privacy guarantee, though it can significantly affect the utility.

However, there is a mismatch between the analysis and implementation of Amin et al. [13]. In their implementation, they form batches of a fixed size $B$ by sampling without replacement. Further, they take the guess of the batch size as $G = B$. Since this depends on the dataset (specifically, via its size $B$), the experimental setting is not covered by their stated DP guarantee (which requires $G$ to be fixed a priori) under add-or-remove adjacency.

We circumvent this issue by choosing an appropriate adjacency notion: we directly analyze the fixed-batch-size version of Amin et al. [13] (i.e. the one that is actually implemented) in the zero-out (Theorem 6) or replace-by-null (Theorem 5) notions of adjacency instead of the add-or-remove

adjacency. As discussed in §B.1, this small change allows us to treat the batch size $B$ as public (see also [15, §2.1.1]), and lets us take $G = B$ without any privacy leakage.

**Sensitivity Bounds.** We have the following bound on the sensitivity of the clipped-and-aggregated logits $\bar{\phi}$:

**Property 9.** *Let the dataset set $B = |\boldsymbol{R}|$ be fixed. Then, the sensitivity of the operation,*[9]

$$f(\boldsymbol{R}) := \frac{1}{|\boldsymbol{R}|} \sum_{i=1}^{|\boldsymbol{R}|} \mathsf{clip}_C\big(\phi(\,\cdot\mid \boldsymbol{q}, \boldsymbol{R}, \boldsymbol{x}_{<t})\big) \tag{8}$$

*is given by:*

$$\mathsf{sens}(f) = \begin{cases} \frac{C}{B}, & \text{under zero-out adjacency}, \\ \frac{2C}{B}, & \text{under replace-by-null adjacency}. \end{cases}$$

*Proof.* We start with zero-out adjacency. Let $\boldsymbol{R} \simeq \boldsymbol{R}'$ differ in their first element where $\boldsymbol{r}'_1 = \perp$. Then, we have that

$$\begin{aligned}
\|f(\boldsymbol{R}) - f(\boldsymbol{R}')\|_\infty &= \frac{1}{B} \left\| \mathsf{clip}_C\big(\phi(\,\cdot\mid \boldsymbol{q}, \boldsymbol{r}_1, \boldsymbol{x}_{<t})\big) - \mathsf{clip}_C\big(\phi(\,\cdot\mid \boldsymbol{q}, \perp, \boldsymbol{x}_{<t})\big) \right\|_\infty \\
&= \frac{1}{B} \left\| \mathsf{clip}_C\big(\phi(\,\cdot\mid \boldsymbol{q}, \boldsymbol{r}_1, \boldsymbol{x}_{<t})\big) \right\|_\infty \\
&\leq \frac{C}{B}.
\end{aligned}$$

Similarly, for replace-by-null adjacency, let $\boldsymbol{R} \simeq \boldsymbol{R}'$ differ in their first element where $\boldsymbol{r}'_1 = \emptyset$ is the empty string.

$$\begin{aligned}
\|f(\boldsymbol{R}) - f(\boldsymbol{R}')\|_\infty &= \frac{1}{B} \left\| \mathsf{clip}_C\big(\phi(\,\cdot\mid \boldsymbol{q}, \boldsymbol{r}_1, \boldsymbol{x}_{<t})\big) - \mathsf{clip}_C\big(\phi(\,\cdot\mid \boldsymbol{q}, \emptyset, \boldsymbol{x}_{<t})\big) \right\|_\infty \\
&\leq \frac{1}{B} \left( \left\| \mathsf{clip}_C\big(\phi(\,\cdot\mid \boldsymbol{q}, \boldsymbol{r}_1, \boldsymbol{x}_{<t})\big) \right\|_\infty + \left\| \mathsf{clip}_C\big(\phi(\,\cdot\mid \boldsymbol{q}, \emptyset, \boldsymbol{x}_{<t})\big) \right\|_\infty \right) \\
&\leq \frac{2C}{B},
\end{aligned}$$

where we used the triangle inequality on the $\ell_\infty$ norm for the first inequality. $\qquad\square$

**Convention in our Experiments.** While we use the more natural replace-by-null adjacency in our experiments, we (incorrectly) use the sensitivity bound obtained for the zero-out adjacency from Theorem 9. This gives Amin et al. [13] an advantage of a factor of 2 in the sensitivity. Yet, we find that INVISIBLEINK outperforms their method in privacy-utility-compute tradeoffs.

### B.3.2 AdaPMixED: Adjacency and Data-Dependent DP Guarantees

AdaPMixED [19] uses the add-remove notion of adjacency, but it gives a data-dependent privacy guarantee, while we give a *data-independent* privacy guarantee under the replace-by-null adjacency.

Strictly speaking, the privacy guarantees across different adjacency notions cannot be directly compared. Thus, we modify AdaPMixED to use the *replace-by-null* adjacency and report these results in the main paper.

We can also cautiously compare the add-or-remove adjacency of AdaPMixED and the replace-by-null adjacency of INVISIBLEINK. This is because the semantics of these two notions of adjacency are expected to be similar [15, 70]. Comparing (AdaPMixED, add-or-remove) and (AdaPMixED, replace-by-null) reveals no significant difference in the qualitative trends; see §E.6 for details.

Furthermore, AdaPMixED gives *data-dependent* privacy guarantees. For example, the zCDP notion (Theorem 3) can be phrased in the data-dependent setting under the *replace-by-null* adjacency as follows. An algorithm $\mathcal{A}(\boldsymbol{q}, \boldsymbol{R})$ satisfies $\rho(\boldsymbol{R})$-zCDP if

$$D_\alpha^{\mathrm{sym}}\big(\mathcal{A}(\boldsymbol{q}, \boldsymbol{R}), \mathcal{A}(\boldsymbol{q}, \boldsymbol{R}')\big) \leq \rho(\boldsymbol{R})\,\alpha$$

---

[9]The clipping function employed by Amin et al. [13] incorporates a recentering step, as logits are invariant to additive shifts. We omit this detail in our exposition as it does not change the mathematical properties.

for all $\alpha > 1$ and all datasets $\boldsymbol{R}'$ that can be obtained from the given dataset $\boldsymbol{R}$ by replacing any one of its elements with the empty string $\emptyset$; here

$$D_\alpha^{\mathrm{sym}}(P, Q) := \max\left\{D_\alpha(P\|Q), D_\alpha(Q\|P)\right\}$$

is symmetrized Rényi divergence. This guarantee holds only for the given sensitive dataset $\boldsymbol{R}$. In contrast, Theorem 3 must hold for all $\boldsymbol{R} \simeq \boldsymbol{R}'$.

Thus, our data-independent guarantees are strictly more general than the data-dependent guarantees given by AdaPMixED. The latter are, in general, privacy-sensitive and need to be sanitized before release. In contrast, our privacy guarantees are worst-case and hold independent of the dataset.

Finally, we have the advantage of straightforward privacy accounting. Given a desired privacy level, compute budget, and maximum generation length (in terms of number of tokens), we give a recipe in §4 to tune the other hyperparameters of our method. In contrast, data-dependent DP guarantees can only be given by grid search on the actual sensitive dataset $\boldsymbol{R}$. In practice, one must account for the privacy cost of such hyperparameter tuning [20], making it potentially more costly in terms of both computation budget and privacy budget.

### B.3.3 Adjacency Notion of AugPE

AugPE [41] is a synthetic text generation algorithm in the API access model. Note that we focus on the less restrictive whitebox model, where per-token model logits are assumed to be available.

AugPE assumes the add-remove notion of adjacency (Theorem 4). Their key technical step is to construct a nearest neighbor histogram. Given a set $S = (\boldsymbol{x}_1, \ldots, \boldsymbol{x}_m)$ of candidate synthetic generations and a set $\mathcal{D} = \{\boldsymbol{r}_1, \ldots, \boldsymbol{r}_N\}$ of real data points, the idea is to compute the nearest neighbor histogram

$$H(\boldsymbol{x}_i) = \sum_{j=1}^N \mathbb{I}(\mathsf{NN}(\boldsymbol{r}_j, S) = \boldsymbol{x}_i) \quad \text{where} \quad \mathsf{NN}(\boldsymbol{r}_j, S) = \operatorname*{arg\,min}_{\boldsymbol{x} \in S} \mathsf{dist}(\boldsymbol{x}, \boldsymbol{r}_j) \tag{9}$$

is the nearest neighbor of $\boldsymbol{r}_j$ from the set $S$ according to a given distance metric $\mathsf{dist}(\cdot, \cdot)$ (with ties resolved arbitrarily) and $\mathbb{I}(\cdot)$ is the indicator function which takes the value 1 when its argument is true and zero otherwise. Xie et al. [41] take $\mathsf{dist}$ as the Euclidean distance in some embedding space. This histogram $H(\cdot) \in \mathbb{R}^m$ is then privatized using the Gaussian mechanism. Its $\ell_2$-sensitivity is as follows (with an elementary proof provided for completeness).

**Property 10.** *For any fixed set of synthetic text $S = \{\boldsymbol{r}_1, \ldots, \boldsymbol{r}_m\}$, the $\ell_2$-sensitivity of the map $\{\boldsymbol{r}_1, \ldots, \boldsymbol{r}_N\} \mapsto (H(\boldsymbol{x}_1), \ldots, H(\boldsymbol{x}_m))$ is:*

- 1 *under the add-or-remove adjacency;*
- $\sqrt{2}$ *under the replace-by-null or zero-out adjacency.*

*Proof.* Under the add-or-remove adjacency, moving from a dataset $\mathcal{D}$ to a neighboring $\mathcal{D}'$ will change at most one entry of the histogram $H(\cdot)$ by at most 1, leading to an $\ell_2$-sensitivity of at most 1. However, under the replace-by-null or zero-out adjacency notions, at most *two* entries of the histogram $H(\cdot)$ will change by at most 1. This leads to an $\ell_2$ sensitivity of $\sqrt{2}$. $\qquad\square$

**Convention in our Experiments.** Similar to the case of Amin et al. [13], we give AugPE an advantage by incorrectly using its best sensitivity from Theorem 10. In particular, we use the more natural replace-by-null adjacency in our experiments but we (incorrectly) use the AugPE's sensitivity bound obtained for the add-or-remove adjacency from Theorem 10. This gives AugPE an advantage of a factor of $\sqrt{2}$ in the sensitivity. Yet, we find that INVISIBLEINK far outperforms AugPE in privacy-utility-compute tradeoffs.

## C   Full Proofs and Additional Details of INVISIBLEINK

Below, we give the proofs of Theorem 1 and Theorem 2. We also give the sensitivity of DClip under different notions of adjacency below.

*Proof of Theorem 1.* Let the dataset size $B = |\boldsymbol{R}|$ be fixed. For replace-by-null adjacency, consider the adjacent dataset $\boldsymbol{R} = \{\boldsymbol{r}_1, \ldots, \boldsymbol{r}_B\}$ and $\boldsymbol{R}' = \{\boldsymbol{r}'_1, \ldots, \boldsymbol{r}'_B\}$ that differ in their first element where $\boldsymbol{r}'_1 = \emptyset$ is the empty string. We need to bound the sensitivity of the map

$$f_{\mathsf{DClip}}(\boldsymbol{r}_1, \ldots, \boldsymbol{r}_B) = \frac{1}{B} \sum_{i=1}^{B} \mathsf{DClip}_C \big( \phi(\cdot \mid \boldsymbol{q}, \boldsymbol{r}_i, \boldsymbol{x}_{<t}), \phi(\cdot \mid \boldsymbol{q}, \boldsymbol{x}_{<t}) \big). \tag{10}$$

Below, we use shorthand $\boldsymbol{\phi}_{\mathsf{pub}} := \phi(\cdot \mid \boldsymbol{q}, \boldsymbol{x}_{<t})$ and $\boldsymbol{\phi}_i := \phi(\cdot \mid \boldsymbol{q}, \boldsymbol{r}_i, \boldsymbol{x}_{<t})$ are public and private logits respectively for $\boldsymbol{R}$ and $\boldsymbol{\phi}'_i := \phi(\cdot \mid \boldsymbol{q}, \boldsymbol{r}'_i, \boldsymbol{x}_{<t})$ are private logits for $\boldsymbol{R}'$. Noting that (a) $\boldsymbol{\phi}_i = \boldsymbol{\phi}'_i$ for $i \neq 1$ by assumption, and (b) $\boldsymbol{\phi}'_1 = \boldsymbol{\phi}_{\mathsf{pub}}$ under the replace-by-null adjacency, we upper bound the sensitivity as

$$
\begin{aligned}
\mathsf{sens}(f_{\mathsf{DClip}}) &= \|f_{\mathsf{DClip}}(\boldsymbol{R}) - f_{\mathsf{DClip}}(\boldsymbol{R}')\|_\infty \\
&= \left\| \frac{1}{B} \sum_{i=1}^{B} \mathsf{clip}_C (\boldsymbol{\phi}_i - \boldsymbol{\phi}_{\mathsf{pub}}) - \frac{1}{B} \sum_{i=1}^{B} \mathsf{clip}_C (\boldsymbol{\phi}'_i - \boldsymbol{\phi}_{\mathsf{pub}}) \right\|_\infty \\
&\stackrel{(a)}{=} \frac{1}{B} \left\| \mathsf{clip}_C (\boldsymbol{\phi}_1 - \boldsymbol{\phi}_{\mathsf{pub}}) - \mathsf{clip}_C (\boldsymbol{\phi}'_1 - \boldsymbol{\phi}_{\mathsf{pub}}) \right\|_\infty \\
&\stackrel{(b)}{=} \frac{1}{B} \left\| \mathsf{clip}_C (\boldsymbol{\phi}_1 - \boldsymbol{\phi}_{\mathsf{pub}}) \right\|_\infty \\
&\leq \frac{C}{B},
\end{aligned}
$$

where the last line follows because $\ell_\infty$ norm of logits clipped using $\mathsf{clip}_C(\cdot)$ is bounded by $C$. $\qquad\square$

*Proof of Theorem 2.* Algorithm 1 consists of the application of the map $f_{\mathsf{DClip}}$, defined previously in Eq. (10), followed by the application of the exponential mechanism with a temperature of $\tau$ (in the form of softmax sampling). Note that the Top-$k+$ selection step in Algorithm 1 is agnostic to the values of the private logits $\boldsymbol{\phi}_1, \ldots, \boldsymbol{\phi}_B$ and therefore does not incur any additional privacy cost.

Consider the generation of one token $x_t$ using Algorithm 1 given arbitrary $\boldsymbol{x}_{<t}$. Using the definition of the exponential mechanism in §B.2 (cf. (7)), the $\rho$-zCDP guarantees for generating any one token $x_t$ is given by

$$\rho_{\mathsf{tok}} = \frac{1}{2} \cdot \left( \frac{\mathsf{sens}(f_{\mathsf{DClip}})}{\tau} \right)^2 = \frac{1}{2} \cdot \left( \frac{C}{B\tau} \right)^2 = \frac{C^2}{2B^2\tau^2}, \tag{11}$$

where $\mathsf{sens}(f_{\mathsf{DClip}})$ is obtained from Theorem 1.

To generate sequences which may contain at most $T$ tokens $x_1, \ldots, x_T$, the zCDP guarantees obtained for single-token generation in Eq. (11) may be adaptively composed sequentially over all $T$ tokens. Then, the entire sequence $\boldsymbol{x} = (x_1, \ldots, x_{T'})$ with $T' \leq T$ satisfies $\rho_{\mathsf{seq}}$-zCDP where (cf. Theorem 13(a))

$$\rho_{\mathsf{seq}} \leq T \cdot \rho_{\mathsf{tok}} = \frac{TC^2}{2B^2\tau^2}.$$

This completes the proof of Theorem 2. $\qquad\square$

Note that this guarantee holds for the generation of multiple sequences using disjoint batches of references from the full sensitive reference dataset by parallel composition of the $\rho$-zCDP guarantees.

**Corollary 11.** *Consider partitioning a dataset $\mathcal{D} = \{\boldsymbol{r}_1, \ldots, \boldsymbol{r}_N\}$ into $n$ disjoint batches $\boldsymbol{R}_1, \ldots, \boldsymbol{R}_n$ of size $B = N/n$ (assumed an integer), i.e., $\boldsymbol{R}_i \cap \boldsymbol{R}_j = \emptyset \, \forall i \neq j$. Suppose further that this partition is data-independent.[10] Suppose we generate texts $\boldsymbol{x}_1, \ldots, \boldsymbol{x}_n$ using Algorithm 1 such that $\boldsymbol{x}_i$ only sees sensitive examples $\boldsymbol{R}_i$ and satisfies $\rho_{\mathsf{seq}}$-zCDP individually. Then, the map $\mathcal{D} \mapsto (\boldsymbol{x}_1, \ldots, \boldsymbol{x}_n)$ also satisfies $\rho_{\mathsf{seq}}$-zCDP.*

*Proof.* This follows from Theorem 2, and from the parallel composition for $\rho$-zCDP guarantees in Theorem 13. $\qquad\square$

---

[10]That is, a partitioning is performed using a process that is agnostic to the content of the data. This ensures that the partition is not influenced by any patterns, information, or outcomes within the dataset. This precludes partitioning based on clustering, for instance.

Note that the privacy guarantee in Theorem 11 does not depend on the number of synthetic texts generated. The number of possible generations using INVISIBLEINK for a fixed dataset size $N$ is thus limited only by the number of possible partitions $n \leq \lfloor N/B \rfloor$.

## C.1 Adapting INVISIBLEINK for other Adjacency Notions

The notion of adjacency used is central to the interpretation of the obtained privacy guarantees, as well as to the sensitivity analysis. We use *replace-by-null* adjacency to derive the privacy guarantees for INVISIBLEINK in Theorem 1 and Theorem 2; refer to §B for a full discussion of other adjacency notions and on how they have been used in previous work.

We also state the sensitivity of $f_{\mathsf{DClip}}$ under *zero-out* adjacency in Theorem 12.

**Property 12.** *Let the dataset set $B = |\boldsymbol{R}|$ be fixed. Then the sensitivity of the map $f_{\mathsf{DClip}}$ : $(\boldsymbol{r}_1, \ldots, \boldsymbol{r}_B) \mapsto \frac{1}{B} \sum_{i=1}^{B} \mathsf{DClip}_C(\boldsymbol{\phi}_i, \boldsymbol{\phi}_{\mathsf{pub}})$ defined in Theorem 1, under zero-out adjacency, is*

$$\mathsf{sens}(f_{\mathsf{DClip}}) = \frac{2C}{B} \,.$$

*Proof.* For zero-out adjacency, consider adjacent datasets $\boldsymbol{R} = \{\boldsymbol{r}_1, \ldots, \boldsymbol{r}_B\}$ and $\boldsymbol{R}' = \{\boldsymbol{r}'_1, \ldots, \boldsymbol{r}'_B\}$ that differ in their first element where $\boldsymbol{r}'_1 = \perp$ is the empty string. We need to bound the sensitivity of the map $f_{\mathsf{DClip}}$ from Eq. (10). As in the proof of Theorem 1, we use shorthand $\boldsymbol{\phi}_{\mathsf{pub}} := \phi(\cdot \,|\, \boldsymbol{q}, \boldsymbol{x}_{<t})$ and $\boldsymbol{\phi}_i := \phi(\cdot \,|\, \boldsymbol{q}, \boldsymbol{r}_i, \boldsymbol{x}_{<t})$ are public and private logits respectively for $\boldsymbol{R}$ and $\boldsymbol{\phi}'_i := \phi(\cdot \,|\, \boldsymbol{q}, \boldsymbol{r}'_i, \boldsymbol{x}_{<t})$ are private logits for $\boldsymbol{R}'$. Noting that (a) $\boldsymbol{\phi}_i = \boldsymbol{\phi}'_i$ for $i \neq 1$ by assumption, and (b) $\boldsymbol{\phi}'_1 = \boldsymbol{0}$ under the zero-out adjacency, we upper bound the sensitivity as

$$
\begin{aligned}
\mathsf{sens}(f_{\mathsf{DClip}}) &= \|f_{\mathsf{DClip}}(\boldsymbol{R}) - f_{\mathsf{DClip}}(\boldsymbol{R}')\|_\infty \\
&= \left\| \frac{1}{B} \sum_{i=1}^{B} \mathsf{clip}_C\big(\boldsymbol{\phi}_i - \boldsymbol{\phi}_{\mathsf{pub}}\big) - \frac{1}{B} \sum_{i=1}^{B} \mathsf{clip}_C\big(\boldsymbol{\phi}'_i - \boldsymbol{\phi}_{\mathsf{pub}}\big) \right\|_\infty \\
&\stackrel{(a)}{=} \frac{1}{B} \left\| \mathsf{clip}_C\big(\boldsymbol{\phi}_1 - \boldsymbol{\phi}_{\mathsf{pub}}\big) - \mathsf{clip}_C\big(\boldsymbol{\phi}'_1 - \boldsymbol{\phi}_{\mathsf{pub}}\big) \right\|_\infty \\
&\stackrel{(b)}{\leq} \frac{1}{B} \left\| \mathsf{clip}_C\big(\boldsymbol{\phi}_1 - \boldsymbol{\phi}_{\mathsf{pub}}\big) - \mathsf{clip}_C\big( - \boldsymbol{\phi}_{\mathsf{pub}}\big) \right\|_\infty \\
&\stackrel{(c)}{\leq} \frac{1}{B} \left\| \mathsf{clip}_C\big(\boldsymbol{\phi}_1 - \boldsymbol{\phi}_{\mathsf{pub}}\big) \right\|_\infty + \frac{1}{B} \left\| \mathsf{clip}_C\big( - \boldsymbol{\phi}_{\mathsf{pub}}\big) \right\|_\infty \\
&\leq \frac{2C}{B} \,,
\end{aligned}
$$

where (c) follows from the triangle inequality on the $\ell_\infty$ norm and the last line follows because the $\ell_\infty$ norm of any logits clipped using $\mathsf{clip}_C(\cdot)$ is upper bounded by $C$. $\qquad \square$

The $2\times$ increase in the sensitivity of $f_{\mathsf{DClip}}$ under *zero-out* adjacency causes a $4\times$ increase in the $\rho$-zCDP guarantee of Algorithm 1 for the same parameters and settings as under the *replace-by-null* adjacency. This highlights the need for careful definition of adjacency notions in DP.

## C.2 Technical Lemmas

We recall some technical results that are useful for other proofs.

**Lemma 13** (Adaptive Composition of zCDP [67])**.** *For any fixed sets $\mathcal{X}, \mathcal{Y}$, let $\mathcal{A}_1 : \mathcal{X}^* \to \mathcal{Y}_1$ be $\rho_1$-zCDP, and $\mathcal{A}_2 : \mathcal{X}^* \times \mathcal{Y}_1 \to \mathcal{Y}_2$ be $\rho_2$-zCDP mechanism with respect to its first argument for any fixed second argument. Then, we have the following adaptive composition results:*

(a) ***Sequential:*** *the mechanism $\mathcal{A}(D) = \big(Y_1, \mathcal{A}_2(D, Y_1)\big)$ for $Y_1 = \mathcal{A}_1(D)$ is $(\rho_1 + \rho_2)$-zCDP.*

(b) ***Parallel:*** *the mechanism $\mathcal{A}(D) = \big(Y_1, \mathcal{A}_2(D_2, Y_1)\big)$ for $Y_1 = \mathcal{A}_1(D_1)$ and any fixed data-independent partition $D_1, D_2$ of $D$ is $\max\{\rho_1, \rho_2\}$-zCDP.*

*This result can be extended to an arbitrary number of mechanisms by induction.*

## C.3 Intuition for Top-$k+$ Sampling

As described in §4, we isolate the contribution of each private logit vector $\phi_i$, following clipping and averaging, as $\phi_i^{\text{clip}}$.

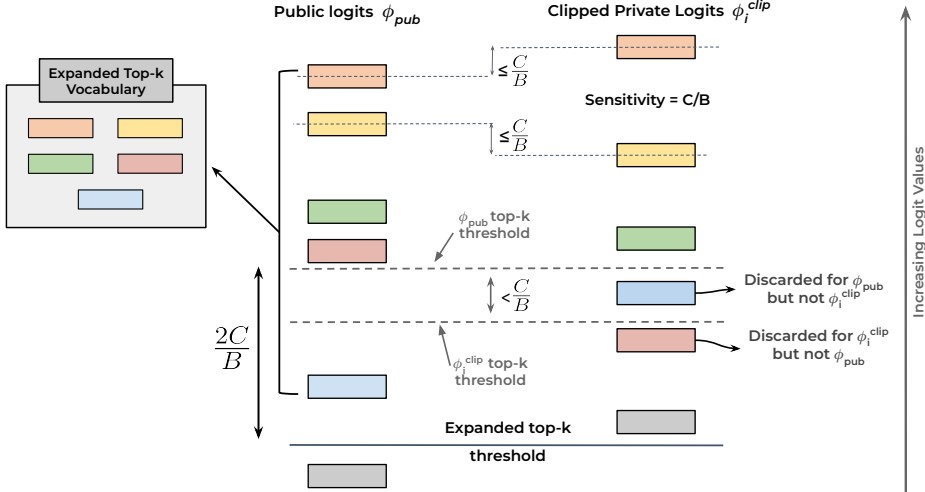

Figure 7: Schematic describing Top-$k+$ sampling. After clipping and aggregating using DClip, the logit values corresponding to each token change by at most $\frac{C}{B}$, i.e., the sensitivity of the private clipped logits. Top-$k+$ sampling, lowers the top-$k$ threshold of the public logits by $\frac{2C}{B}$. The expanded vocabulary $V_k^+$ captures all tokens in a superset of the top-$k$ vocabularies of the public logits $\phi_{\text{pub}}$, and all the clipped private logits $\phi_i^{\text{clip}}$.

In Fig. 7, the logit values of each token in $\phi_i^{\text{clip}}$ differ from those in $\phi_{\text{pub}}$ by at most $\frac{C}{B}$. Further, the top-$k$ sets of both these logits may differ as token ordering around the top-$k$ threshold changes in the following two ways: (i) tokens in the top-$k$ set of $\phi_{\text{pub}}$ can drop out of the top-$k$ set of $\phi_i^{\text{clip}}$ as their logit value decreases, and (ii) tokens outside the top-$k$ set of $\phi_{\text{pub}}$ can enter the top-$k$ set of $\phi_i^{\text{clip}}$ as their logit value increases.

All tokens which exit the top-$k$ set of $\phi_{\text{pub}}$ due to (i), must still lie within $\frac{C}{B}$ of the top-$k$ threshold of $\phi_{\text{pub}}$. However, this could potentially decrease the top-$k$ threshold of $\phi_{\text{pub}}$ by at most $\frac{C}{B}$ when moving to $\phi_i^{\text{clip}}$. Thus, all tokens which enter the top-$k$ set of $\phi_i^{\text{clip}}$ due to (ii) must lie within $\frac{2C}{B}$ of the top-$k$ threshold of $\phi_{\text{pub}}$. Thus, lowering the top-$k$ threshold of $\phi_{\text{pub}}$ by $\frac{2C}{B}$ captures all tokens of interest, as shown by the toy example in Fig. 7.

# D   Experimental Setup

## D.1   Task: Synthetic Text Generation

Our goal is to generate high-quality, long-form synthetic data from a given private dataset of references, while under rigorous DP guarantees on the privacy of the references. We model the references $\boldsymbol{R} = (\boldsymbol{r}_1, \ldots, \boldsymbol{r}_{|B|})$ as part of the context for next-token generation in the LLM. The query prompts $\boldsymbol{q}$, contain instructions for text generation including general details about the sensitive dataset such as domain, type of text document, etc. The public and private (here, private prompts mean prompts with sensitive references embedded in-context) query prompts used for the generation of synthetic text samples for various datasets are listed in Tab. 4. Examples of the references for both Yelp and TAB datasets are given in Tab. 5.

Algorithm 1 describes the generation of 1 text sample from a batch of sensitive references $\boldsymbol{R}$ of size $B$. To generate multiple synthetic text samples, we partition the entirety of the sensitive reference dataset into $B$-sized chunks. Since the dataset partitions are defined in a completely data-independent manner, we may compose the DP guarantee parallely across batches; see Theorem 11. The theoretical

privacy guarantee for the generation of multiple text samples is thus the same as that of 1 text sample, i.e., the overall privacy guarantees are independent of the number of synthetic samples generated and only depend on the maximum number of BPE encoded tokens in each input.

| Dataset | Prompt Type | Prompt Text |
|---|---|---|
| **MIMIC** | **Private** | Here is the text of the discharge summary of a patient discharged from a hospital
Text: "*<private reference here>*"
Please give me another text of a fake patient discharge summary for discharge from a hospital. Include typical sections like admitting diagnosis, major procedures (if any), discharge medications (using fictional drug names and dosages), and general follow-up instructions. Do not include any names, dates, or specific medical record numbers. The output text must begin with the exact words "Discharge Instructions:". |
| | **Public** | Please give me text of a fake patient discharge summary for discharge from a hospital. I only need fictional and representative examples for a non-medical purpose. Include typical sections like admitting diagnosis, major procedures (if any), discharge medications (using fictional drug names and dosages), and general follow-up instructions. Do not include any names, dates, or specific medical record numbers. The output text must begin with the exact words "Discharge Instructions:". |
| **Yelp** | **Private** | Here is a text with Business Category : "*<business category>*" and Review Stars : "*<review score>*" out of 5.0.
Text: "*<private reference here>*"
Please output one more similar review of a fictional place with the same score and of the same category of business. Poor reviews have lower scores and good reviews have higher scores. |
| | **Public** | Please give me a fake Yelp customer review of a fictional business with Business Category : "*<business category>*" and Review Stars : "*<review score>*" out of 5.0. Poor reviews have lower scores and good reviews have higher scores. |
| **TAB** | **Private** | Here is the text of a case transcript set before the European Court for Human Rights.
Text: "*<private reference here>*"
Please output a similar transcript of a fictional case under European Court for Human Rights. Begin with the phrase: 'PROCEDURE: The case originated in an application'. |
| | **Public** | Please output a transcript of a fictional case under European Court for Human Rights. Begin with the phrase: 'PROCEDURE: The case originated in an application'. |

Table 4: Public and private prompts for MIMIC, Yelp and TAB datasets; The private references are inserted in the space denoted by "*<private reference here>*". The system prompt across all settings is: "You are a chatbot".

## D.2 Datasets

We use both publicly available open-access datasets and licensed datasets: MIMIC IV Clinical Notes dataset [78, 79], Yelp Reviews dataset [81], and the Text Anonymization Benchmark (TAB) [80] dataset. The respective license and data use agreements of the various assets we use have been duly followed throughout this work.

**MIMIC-IV Clinical Notes.** This dataset is a large, de-identified collection of medical text associated with ICU patients, including discharge summaries, radiology reports, and nursing notes. It is widely used for clinical NLP tasks such as diagnosis classification, named entity recognition, and patient cohort identification. The dataset poses significant privacy challenges due to the presence of sensitive patient information, making it an ideal choice for evaluating privacy-preserving language models.

| Dataset | Sensitive References (Exemplars) |
| --- | --- |
| Yelp | I've been here a few times. The service has been very good all times. The food is hit or miss on some entrees. Do not order the poutine fries. They have no sauce. It's like eating fries with cheese and meat on them. No gravy like poutine is supposed to have. Their salmon special that they had one day was very good. It had an Asian flare. I also have had the Asian noodle dish which was good. Their burgers are vet popular there and look tasty. There is something on the menu for everyone! They Also have good cocktails. |
| TAB | PROCEDURE The case originated in an application (no. 36244/06) against the Kingdom of Denmark lodged with the Court under Article 34 of the Convention for the Protection of Human Rights and Fundamental Freedoms ("the Convention") by a Danish national, Mr Henrik Hasslund ("the applicant"), on 31 August 2006. The applicant was represented by Mr Tyge Trier, a lawyer practising in Copenhagen. The Danish Government ("the Government") were represented by their Agent, Ms Nina Holst-Christensen of the Ministry of Justice. On 5 September 2007 the Acting President of the Fifth Section decided to give notice of the application to the Government. It was also decided to rule on the admissibility and merits of the application at the same time (Article 29 §3). |

Table 5: Examples from the Yelp and TAB datasets. We use these are privacy-sensitive references for synthetic text generation. We do not provide examples of the MIMIC dataset (or even synthetic generations) to comply with its license.

Further, the dataset is not available for open-access and cannot legally be used for pre-training general-purpose LLMs such as those we use in our experiments. This dataset is released under the PhysioNet Credentialed Health Data License 1.5.0, and may be accessed via the PhysioNet website (https://physionet.org/content/mimic-iv-note/2.2/).

For our synthetic text generation task, we use the discharge summaries released as part of the dataset. In particular, we extract the discharge instructions from within these files. These are selected as they often contain succinct summaries of patient symptoms, diagnosis, treatment, as well as aftercare instructions. Further, almost all discharge summaries contain a section on "Discharge Instruction" allowing us to extract almost 311K valid private references from the dataset. In accordance with the License and Data Use Agreement of the dataset, we do not release the original dataset, processed text samples, or synthetically generated private samples.

**Yelp Reviews.** We use the processed Yelp Open dataset [81] from [44] released by [41]. This dataset contains 1.9M of user-generated reviews and ratings for businesses (see Tab. 1 for a precise size), labelled by business category (10 categories) and review scores (5 possible scores) to give a total of 50 sub-categories. Although not overtly sensitive, it contains personal opinions and location references and has previously been used to design private synthetic text generation algorithms [41, 44].

The Yelp Open dataset is released under a data agreement and terms of use, which are present in the downloadable .zip file hosted by Yelp. This is distinct from but similar to the (also publicly available) Yelp Reviews dataset hosted by HuggingFace; in particular, they contain examples of similar reviews, labelled by review scores across several (unlabelled) business categories.

The respective GitHub repositories of [41] (link) and [44] (link) are released under the Apache 2.0 and MIT licenses. The license under which the processed dataset is released is not explicitly mentioned in either repository. The dataset may be downloaded from this link by following the instructions given by [41].

**Text Anonymization Benchmark (TAB).** This dataset [80] is a dedicated, open-source corpus designed for evaluating text anonymization systems. It comprises 1268 English-language court cases from the European Court of Human Rights (ECHR), manually annotated with semantic categories of personal identifiers, masking decisions based on re-identification risk, confidential attributes, and co-reference relations. TAB provides a high-quality benchmark for assessing the effectiveness of anonymization models in legal and privacy-critical domains. We use the text released in this dataset as a low-resource task for generating high-quality synthetic data when the batch size is constrained to be small. Using a large batch size for this dataset results in a very small yield of synthetic data, for example: $B = 127$ gives us a maximum of 8 synthetic examples — this constrains us to use a batch size $B \leq 7$ if we wish to generate at least 100 examples.

The dataset is released under the MIT License and is publicly available at this link.

**Note on Choice of Dataset.** We note that several common datasets used in prior work (such as AGNews, TREC, DBPedia, WikiMovies, etc.) are incompatible with our setting of long-form text generation due to their relatively shorter texts ($\leq 200$ tokens). Further, several of these datasets (like IMDb, etc.) are well-represented in the pretraining data of most LLMs. Since publicly prompted LLMs can generate high-quality synthetic text for such datasets, a high degree of overlap between the pretraining and target data distributions skews the conclusions of private data analysis [82]. Thus, we prefer using benchmarks that are sufficiently out-of-distribution relative to the pretraining data and preferably contain real privacy-sensitive data.

Accordingly, we use the MIMIC IV Notes dataset as the primary dataset for experiments, since it contains real patient discharge notes with sensitive medical data, and cannot legally be used for pretraining LLMs as per its data use agreement. Similarly, the TAB dataset contains sensitive identifying information (demographic traits, spatio-temporal markers, biographical details, etc.) from European Court of Human Rights cases. We use the Yelp dataset as a representative dataset that overlaps significantly with the pretraining data (similar to the other datasets discussed above).

## D.3 Models

We use the following language models in our empirical evaluation: TinyLLaMA 1B [84], LLaMA3.2-1B [85], and LLaMA3 8B [86]. All models were loaded in `torch.bfloat16` precision.

TinyLLaMA is a compact, open-weight language model trained on 1 trillion tokens and with only 1.1B parameters. Despite a smaller size, it retains strong performance on basic language modeling tasks and is ideal for fast experimentation in compute-constrained settings. For our experiments, we used the instruction-tuned version `TinyLlama-1.1B-Chat-v1.0` of this model. This model is released under the Apache 2.0 License and may be accessed via HuggingFace at: TinyLLaMA 1B.

The LLaMA3.2 1B model is a smaller variant from the LLaMA3 family. Being a multilingual model with a very large vocabulary ($128K$ tokens) and context size, it is an ideal candidate to demonstrate the improvements offered by Top-$k+$ sampling. Additionally, we also conduct one set of experiments on a larger LLaMA3 8B model. For our experiments, we use instruction-tuned versions of both models. The former is released under the Llama 3.2 Community License Agreement and may be accessed via huggingface at: LLaMA3.2 1B. The latter is released under the Llama 3 Community License Agreement and may be accessed via huggingface at: LLaMA3 8B.

## D.4 Evaluation Metrics

For our empirical evaluation, we evaluate the generated text on several different metrics to highlight the advantages of INVISIBLEINK compared to baselines.

**MAUVE score.** To evaluate the distributional similarity between synthetic and real text corpora, we use the MAUVE score [17, 18, 87], which is designed to capture both quality and diversity in text generation quality. Unlike traditional metrics (e.g., BLEU, ROUGE), MAUVE compares entire distributions rather than individual samples. The result is a number between 0 and 1, with 1 denoting that the two distributions are identical, and 0 denoting that the two distributions have no similarity.

We make the following choices to compute the MAUVE scores:

- Implementation: We use the open-source `mauve-text` package released by [17, 18].[11]
- Reference samples: Held-out samples used
- Number of generations and references: Varying between 100 and 1000, as described in Tab. 1. We use an equal number of samples from both the generations and references.
- Embedding model: Sentence Transformer model [92], version `all-mpnet-base-v2`.[12]
- Number of bins: $N/20$ where both reference and generated distributions have $N$ samples.
- Scaling factor: 0.9, across all settings throughout this work.

To compute all MAUVE scores in this paper, we use the SentenceTransformer embeddings of the text sample [92]. We find these embeddings to be of higher quality, and corresponding to better and

---

[11]Available at `https://github.com/krishnap25/mauve`.
[12]Available at `https://huggingface.co/sentence-transformers/all-mpnet-base-v2`.

| MedNER Category | Example(s) | Counted |
|---|---|---|
| THERAPEUTIC_PROCEDURE | "paracentesis", "colonoscopy" | ✓ |
| DIAGNOSTIC_PROCEDURE | "blood pressure", "endoscopy" | ✓ |
| DETAILED_DESCRIPTION | "IV", "constipating" | ✓ |
| DISEASE_DISORDER | "cirrhosis", "pneumonia" | ✓ |
| SIGN_SYMPTOM | "pain", "fever" | ✓ |
| SEVERITY | "worsening", "severe" | ✓ |
| DISTANCE | "1.5L", "5 pounds" | ✓ |
| DOSAGE | "40 mg", "1 tablet every 6 hours" | ✓ |
| LAB_VALUE | "high", "improved" | ✓ |
| MEDICATION | "Lasix", "Tylenol" | ✓ |
| COREFERENCE | "incision", "splint" | ✓ |
| ADMINISTRATION | "intravenous", "oral" | ✓ |
| BIOLOGICAL_STRUCTURE | "stomach", "esophagus" | ✓ |
| NONBIOLOGICAL_LOCATION | "hospital", "emergency room" | × |
| FAMILY_HISTORY$^*$ | "yourself" | × |
| CLINICAL_EVENT | "admitted", "discharged" | × |
| DURATION | "1 day", "several weeks" | × |
| HISTORY | "alcohol", "activities" | × |
| DATE | "1 week", "24 hours" | × |
| AGE$^*$ | "up" | × |
| SEX | "male", "female" | × |
| TIME | ":55pm", ":20am" | × |
| AREA$^*$ | "." | × |
| VOLUME | "7 x 6.7 x 7.6 cm", "4.9 x 6.4 centimeter" | × |
| OTHER_ENTITY | "2", "day" | × |

Table 6: **Medical Named Entities**: The various medical named entities captured by the DeBERTa-Med-NER model [88] and some identified examples for each category. In our MedNER evaluations, we only count the most medically relevant among these categories, as denoted by the rightmost column. Note that some of the omitted columns are not reliably detected by the model (denoted by $^*$).

more meaningful MAUVE scores. In particular, we use the `all-mpnet-base-v2` model, which returns 768-dimensional embeddings of input texts, truncated to 384 tokens. The final score is also scaled appropriately to facilitate easy comparisons by setting the scaling factor to 0.9. It is crucial to note that the value of the scaling factor and other hyperparameters used to calculate MAUVE scores impact the obtained scores. MAUVE scores are thus best used for relative comparisons, such as in this work, and the absolute values themselves are less meaningful.

**Medical Named-Entity Recognition (MedNER).** The original MIMIC discharge notes contain details on various medically significant terms such as diseases, medications, treatments, etc. We found in our early experiments that some generation methods can fail to generate such domain-specific terms. Since we wish the synthetic generations to reflect all statistical properties of the original dataset, it is desirable for generation algorithms to also generate relevant medical entities, at the same rate as the original data.

In other words, we are interested in Medical Named Entity Recognition (Medical NER). This is a common task that focuses on identifying clinically significant entities such as diseases, medications, procedures, anatomical terms, and patient attributes from clinical notes, health records, or other medically relevant text data. We use the counts of medical named-entities in the generated text as a measure of the level of detail and specificity preserved in the synthetic text.

We use the *DeBERTa-Med-NER-2* [88] model to label all medical named entities in the generated text. The model recognizes 41 categories of named-entities of which the text subset of the MIMIC dataset we extract, as described in §D.2, contains only 25 classes. We further select and count the presence of the most relevant medical named entities; 13 classes like therapeutic procedure and disease/disorder are included, but domain-agnostic classes like date and time are excluded. The exact details are presented in Tab. 6.

**Generation Perplexity.** Perplexity is a standard metric used to evaluate the quality of generations of probabilistic language models, including LLMs. Perplexity is defined as the exponent of the average negative log-likelihood of generating every token of the observed sequence, conditioned on

the previously generated tokens. The perplexity of a sequence $\boldsymbol{x} = (x_1, x_2, \ldots)$ is defined as

$$\text{PPL}(\boldsymbol{x}) = \exp\left(-\frac{1}{|\boldsymbol{x}|}\sum_{t=1}^{|\boldsymbol{x}|}\log\hat{P}(x_t \mid \boldsymbol{x}_{<t})\right)$$

where $\log\hat{P}(x_t \mid \boldsymbol{x}_{<t})$ is the log-likelihood of the $t^{\text{th}}$ token conditioned on the preceding tokens $\boldsymbol{x}_{<t}$ (with $\boldsymbol{x}_{<1} = \emptyset$), according to the language model $\hat{P}$. We take the model $\hat{P}$ used to calculate the perplexity of all text samples as GPT-2 small (with $117M$ parameters).

A low perplexity value indicates that the model assigns a higher likelihood to the observed sequence. We wish the generated synthetic text to match the original text dataset in all statistical properties, including perplexity under any given language model. Thus, achieving a small difference in the average perplexities between the generated and original text distributions is desirable. This is measured by the $\Delta$PPL metric [16], defined as

$$\Delta\text{PPL} = \left|\frac{1}{|\boldsymbol{X}|}\sum_{\boldsymbol{x}\in\boldsymbol{X}}\text{PPL}(\boldsymbol{x}) - \frac{1}{|\boldsymbol{R}|}\sum_{\boldsymbol{r}\in\boldsymbol{R}}\text{PPL}(\boldsymbol{r})\right|,$$

where $\boldsymbol{R}$ is the original dataset of sensitive references and $\boldsymbol{X}$ is the synthetically generated dataset.

**Length Distributions.** The privacy budget of per-token privatization algorithms is composed over the number of tokens generated. For methods like Amin et al. [13] and AdaPMixED [19], the number of tokens generated is limited by the privacy budget. In contrast, for our method, we set a predetermined maximum number of allowed tokens and calculate other hyperparameters based on this number. Further, unlike previous methods that are not well-suited for long-form text generation due to either computational or privacy constraints, we aim to produce longer and higher-quality text. Since the length of generated text is another statistical property of the private reference dataset we wish to mimic, we use the number of tokens generated by the model, referred to henceforth as the generation length, as an evaluation metric.

The set of all possible tokens, called the vocabulary, differs from model to model. A tokenizer is used by language models to convert text strings into sequences of valid tokens (each denoted by an integer). The LLaMA3 models use a byte-pair encoding (BPE) model based on tiktoken, whereas TinyLLaMA uses a BPE model based on SentencePiece. A characteristic of BPE models, introduced in [93], is sub-word tokenization, which splits up one word into one or more tokens depending on the length of the word and the substrings in it. The number of words in a text sample is thus not equal to the number of tokens in the same sample. The latter also depends on the exact tokenizer used.

To measure the generation length for generations corresponding to a given model, we use the number of BPE tokens generated by the corresponding tokenizer. Generation lengths reported are thus not directly comparable across families of models. The generation lengths of real texts are calculated separately for every model and reported with the corresponding results (wherever applicable) for comparison. We report the average generation length (in number of tokens) over all generations of $\boldsymbol{x}$ (and its confidence intervals, as described in §D.6).

## D.5 Hyperparameter Selection

We discuss the choices of hyperparameters, the range explored, and the settings for reported results across all methods, INVISIBLEINK and other baselines, in this section. We also note that to allow comparisons between methods, we convert all DP guarantees into $(\varepsilon, \delta)$ guarantees[13].

**INVISIBLEINK.** INVISIBLEINK, described in Algorithm 1, has the following hyperparameters: the clipping norm $C$, the batch size $B$, the sampling temperature $\tau$ and the top-$k$ sampling parameter. For a given $\varepsilon$, the heuristics for choosing optimal hyperparameters are given in §5. To arrive at this heuristic, we conducted a thorough hyperparameter search for a full vocabulary $k = |V|$ variant of the model; the full range of hyperparameters explored for each dataset and model we use is given in Tab. 7. We report results by tuning the sampling temperature for various values of $\varepsilon$ and $B$.

Based on these results (presented in detail in §E) and on the exploration of non-private generation in Fig. 2, we report results for INVISIBLEINK with a fixed sampling temperature $\tau = 1.1$ for $k < |V|$

---

[13]$\delta$ for all methods is chosen to be a sufficiently small value of $10^{-6}$.

| Dataset | Model | Hyperparameter | Range Explored |
|---------|-------|----------------|----------------|
| **MIMIC** | TinyLLaMA 1B | $B$ $\tau$ | $[4, 8, 16, 32]$ $[0.8, 0.9, 1.0, 1.1, 1.2]$ |
| | LLaMA3.2 1B | $B$ $\tau$ | 8 1.1 |
| | LLaMA3 8B | $B$ $\tau$ | 8 1.2 |
| **Yelp** | TinyLLaMA 1B | $B$ $\tau$ | $[2, 4, 8, 16, 32]$ $[0.8, 1.0, 1.2]$ |
| **TAB** | TinyLLaMA 1B | $B$ $\tau$ | 8 $[0.8, 1.0, 1.2]$ |

Table 7: Hyperparameters for INVISIBLEINK variant with $k = |V|$.

and $\tau = 1.0$ for $k = |V|$. We explore a range of top-$k$ parameters for various settings; see Tab. 8. Our empirical results in §5 show that setting $k = 50 - 100$ shows superior performance to all existing baselines, as well as the full-vocabulary ($k = |V|$) variant of INVISIBLEINK.

| Dataset | Model | Hyperparameter | Range Explored |
|---------|-------|----------------|----------------|
| **MIMIC** | TinyLLaMA 1B | $B$ $k$ $\tau$ | $[4, 8, 16, 32]$ $[10, 50, 100, 500]$ $[1.0, 1.1., 1.2]$ |
| | LLaMA3.2 1B | $B$ $k$ $\tau$ | 8 $[10, 50, 100, 500]$ 1.1 |
| | LLaMA3 8B | $B$ $k$ | 8 $[10, 50, 100, 500, 1000, 5000]$ |
| **Yelp** | TinyLLaMA 1B | $B$ $k$ | $[4, 8, 16, 32]$ $[10, 50, 100]$ |
| **TAB** | TinyLLaMA 1B | $B$ $k$ | 8 $[10, 50, 100]$ |

Table 8: Hyperparameters for INVISIBLEINK ($k < |V|$). We set $\tau = 1.2$ wherever unspecified.

**Amin et al. [13].** The algorithm proposed by [13] has similar hyperparameters: $C$ clipping norm, $B$ batch size, $\tau_{prv}$ private sampling temperature, $\tau_{pub}$ public sampling temperature, $\theta$ threshold for Sparse Vector Technique (SVT), $\sigma$ noise parameter for SVT. Further, the privacy guarantees depend on the number of times a token is sampled from the aggregated and clipped private logits. As

| Dataset | Model | Hyperparameter | Range Explored |
|---------|-------|----------------|----------------|
| **MIMIC** | TinyLLaMA 1B | $B$ $\tau$ | $[32, 64, 128]$ $[0.8, 1.0, 1.2]$ |
| **Yelp** | TinyLLaMA 1B | $B$ $\tau$ | $[32, 64, 128]$ $[0.8, 1.0, 1.2]$ |

Table 9: Hyperparameters for Amin et al. [13]'s method.

mentioned in §5, preliminary experimentation showed that the SVT step in [13]'s method selected private tokens $\approx 25\%$ of the time. We select a value of $T_{prv} =$100 for all datasets (where $T_{prv}$ is the maximum number of private tokens allowed) and maintain the total tokens generated at the same levels as the settings for INVISIBLEINK described in Tab. 1. As seen in the plots in Fig. 4, this does not affect the average generation length of synthetic samples generated by this method. As seen in §5, for a number of settings, this method fails to yield any synthetic text (INF or infeasible in Tab. 2). All results available are reported tuned on the sampling temperature $\tau$.

**AdaPMixED [19].** AdaPMixED [19] is a next-token prediction algorithm adapted to generate synthetic text sequences sampling one token at a time under data-dependent privacy guarantees. The

approach described in [19] uses models fine-tuned (without privacy) on partitions of the private references dataset. We focus on private inference methods and adapt AdaPMixED in a fashion similar to [13], using the sensitive references as context for the synthetic text generation. Private inference can be used on pretrained models prompted with private data (our setting) or non-DP fine-tuned models. Both are conceptually identical as they privatize a set of non-private predictions.

We use a custom implementation of the method, modifying it to use the *replace-by-null* adjacency; see §B.3 & §E.6. We note that AdaPMixED has significant (non-GPU) computational overhead compared to other methods due to repeated calculation of optimal mixing parameters; see [19, Alg 1, Line 11]. We implement this using the `brentq` optimizer from the `scipy` Python package. For generations, we use the same values of $T$ (maximum allowed tokens per generation) as discussed previously; see Tab. 1. We fix some hyperparameters across settings: sampling temperature at $\tau = 1$, noise multiplier at $\sigma = 0.01$ and mixing parameter for noisy screening $\lambda = 10^{-4}$; for all other hyperparameters, we use the recommended default values for the PubMed dataset in [19, Tab 4].

**AugPE [41].** AugPE is a private text generation algorithm that relies on iterative LLM-based paraphrasing and construction of privatized (noisy) nearest neighbour histograms at every step to generate private text. See §E.1 for a full discussion on all the generation settings for this method.

### D.6 Confidence Intervals

We report the mean of the evaluation metrics across all generated text samples for a given generation setting, wherever applicable. In addition, we also report the $(1 - \alpha) \times 100\% = 99\%$ confidence intervals calculated using the standard error (standard deviation of the mean) as follows,

$$\mathsf{CI} = \bar{x} \pm Z_{\alpha/2} \frac{s}{\sqrt{m}},$$

where $\bar{x}$ is the sample mean, $s$ is the sample standard deviation and $m$ is the number of samples. Here, $Z_{\alpha/2}$ is the $(1 - \alpha/2)$ percentile of the standard normal distribution. These are Wald confidence intervals, given under the assumption that error is unbiased and normally distributed. For a $99\%$ confidence interval, we have $Z_{\alpha/2} = 2.576$ and for a $95\%$ confidence interval, we have $Z_{\alpha/2} = 1.96$.

### D.7 Computational Resources

**Software.** We used Python 3.13.2, PyTorch 2.6 and HuggingFace Transformers 4.50.1.

**Hardware.** All experiments requiring a GPU (LLM inferences, $\Delta$PPL calculation, MedNER count calculation etc.) were conducted on one of two machines: (a) with 4 NVIDIA RTX L40S GPUs (with 48G memory each), or (b) with 4 NVIDIA H100 GPUs (with 80GB memory each). Each task used only one GPU at a time. All wall-clock time measurements, if any, are performed using the L40S GPUs. The non-GPU jobs, such as computing MAUVE, were run on a machine with 240 AMD EPYC 9554 CPUs (clock speed: 3.10 GHz) with 64 virtual cores each and total memory of 845GB.

**Evaluation Time.** The approximate wall clock runtimes for synthetic text generation using various methods are given below. The runtime varies with the size of the model and batch size used, in particular with the maximum number of parallel LLM inferences that may be run on the GPU. For our experiments using TinyLLaMA, a maximum of $min(B + 1, 16)$ inferences were run in parallel on 1 GPU. The runtimes also scale (approximately) linearly with the length of individual generated samples and the number of samples generated. All numbers reported below are for 1000 generations of 500 token synthetic samples using MIMIC as the reference dataset.

- INVISIBLEINK: $\approx 4$ hours for $B = 15$
- Amin et al. [13]: $\approx 12$ hours for $B = 127$
- AdaPMixED [19]: $\approx 12$ hours for $B = 127$ and $\varepsilon = 1$
- AugPE [41]: $\approx 18$ hours for 7 variations per iteration and 10 iterations

The runtimes for AdaPMixED [19], in particular, have a very high variance with both $B$ and $\varepsilon$ as a higher privacy budget corresponds to longer generations and higher runtimes. For $B = 127$ and $\varepsilon \geq 5$, the runtime exceeds 24 hours for 1000 generations, denoted by TLE in our experimental results. The runtimes for AugPE [41] scale (approximately) linearly with both the number of iterations and the number of variations generated per iteration.

# E   Additional Experimental Results

In this section, we present the following additional results:

- Additional results comparing INVISIBLEINK and AugPE [41] on the MIMIC dataset.
- Additional experiments on the downstream utility of data generated from the Yelp dataset.
- Additional statistics for the selection of tokens from the expanded vocabulary described by the Top-$k+$ step in INVISIBLEINK.
- Additional results for synthetic generations for the MIMIC dataset using LLaMA3 8B.
- Additional experiments designed to highlight the effect of sampling temperature $\tau$, clipping norm $C$ and top-$k$ sampling parameter on the quality of generation.
- Experiments comparing the quality of private text generation for AdaPMixED with the replace-by-null adjacency and add-or-remove notions of adjacency.

## E.1   Comparison with AugPE

Fig. 8 gives additional experimental results comparing the utility vs computational cost of INVISIBLEINK and AugPE for various privacy budgets. The trends agree with those in Fig. 5; see §5.

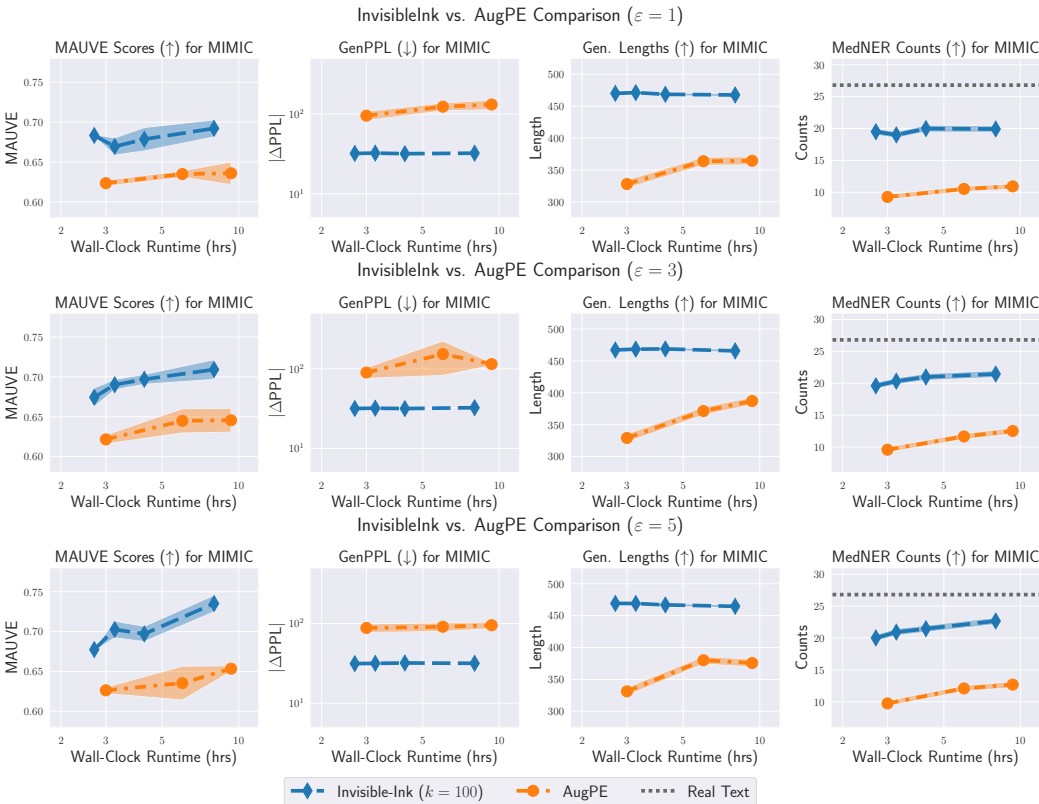

Figure 8: Utility vs Compute plots (avg. over 3 runs) for INVISIBLEINK and AugPE for varying privacy budgets $\varepsilon$ for 1000 synthetic texts generated for the MIMIC dataset. We compare utility using a variety of metrics — INVISIBLEINK outperforms AugPE across all settings and evaluation metrics. Wall-clock run time is used to measure the computational cost. We report results for $B + 1 = 4, 8, 16, 32$ next-token inference calls per generated token and $T_{\mathsf{AugPE}} = 1, 3, 5$ for INVISIBLEINK and AugPE, respectively.

The number of variations of the candidate synthetic texts generated in every iteration of AugPE (the parameter $L$ in [41, Alg 1]) is fixed to be 7. We also limit the total number of sensitive references the algorithm sees to $7 \times n$ (where $n$ is the total number of synthetic texts generated). The total number of full-sequence LLM generations per synthetic text is given by $8 \times T_{\mathsf{AugPE}}$, where $T_{\mathsf{AugPE}}$ is the

| Prompt Type | Prompt Text |
|---|---|
| **System Prompt** | Please act as a sentence generator for the medical domain. Generated sentences should mimic the style of medical discharge summaries, using a variety of sentence structures. |
| **Generation Prompt** | Using a variety of sentence structures, write a patient discharge summary. Begin the output with the phrase: "Discharge Instructions:" |
| **Variation Prompt** | Please rephrase the following sentences "*<insert candidate synthetic text>*" as a patient discharge summary: "*<insert rephrase style>*"
Begin the output with the phrase: "Discharge Instructions:" |

Table 10: Random sample generation and Variation prompts for MIMIC adapted in the style of AugPE; The candidate synthetic generations at the current iteration are inserted in the space denoted by "*<insert candidate synthetic text>*" and information about the style in which the LLM should rephrase the text is inserted in the space denoted by "*<insert rephrase style>*".

number of iterations, and varies from $8 - 80$. In contrast, we disadvantage our model by limiting the number of effective full-sequence LLM generations to $B + 1$, ie, $4 - 32$. Further, we use the official implementation of AugPE released by Xie et al. [41][14] and adapt it for generating synthetic samples from MIMIC with minimal changes. In particular, we preserve the same style of random sample generation and text variation prompts used for other datasets; see Tab. 10.

### E.2 Yelp Downstream Utility Experiments

We evaluate the synthetic data generated from the Yelp dataset on a downstream classification task. We finetune a RoBERTa model [94] for sequence classification on a dataset of $500$ generated synthetic texts from the Yelp dataset into $50$ classes, obtained as all combinations of $5$ review score labels and $10$ business category labels. We note that since the synthetic data generation is conditioned on these classes, the class label information is also available in the synthetic data.

We evaluate the resulting classifier on the Yelp test set. Tab. 11 reports the 50-class top-1 and top-5 accuracy (resp. columns "Accuracy" and "Top-5 Acc."), 10-class accuracy over categories (column "Cat. Acc."), 5-class accuracy over score labels (column "Score Acc."), and the $L_1$ distance between the predicted and true numeric scores (column "Score $L_1$"). We finetune the RoBERTa model [94] for $15$ epochs with an AdamW optimizer [95] with a batch size of $20$, a learning rate of $2 \times 10^{-5}$, a weight decay of $0.01$ and $10\%$ warmup steps.

The results presented in Tab. 11 compare the downstream utility metrics for text generated by various methods with a fixed privacy budget of $\varepsilon = 10$. INVISIBLEINK outperforms all baseline methods by a large margin at a significantly smaller computation cost, even when measured in terms of this downstream task utility.

| Method | Batch Size $B$ | Accuracy (↑) | Top-5 Acc. (↑) | Cat. Acc. (↑) | Score Acc. (↑) | Score $L_1$ (↓) |
|---|---|---|---|---|---|---|
| INVISIBLEINK | 7 | **32.98** | **72.16** | **64.90** | **52.2** | **0.652** |
| Amin et al. [13] | 127 | 29.44 | 64.56 | 60.18 | 46.64 | 0.748 |
| AdaPMixED [19] | 31 | 7.44 | 34.72 | 56.82 | 15.44 | 1.858 |

Table 11: **Downstream utility of synthetic text** Multiple evaluation metrics task for a sequence classification task using a RoBERTa [94] model finetuned on synthetic sequences generated from the Yelp dataset with privacy budget $\varepsilon = 10$. INVISIBLEINK outperforms the baselines significantly.

We note that measuring utility in downstream tasks is not a reliable indicator of overall text quality, coherence, or distributional similarity to the original dataset. Several prior works on DP text generation [13, 19, 47, 48] rely heavily on such metrics to evaluate generated text. However, it is possible for poor-quality synthetic text to "hack" these metrics, especially in the case of sequence classification by relying on keywords related to the target category. This can be observed in the case of AdaPMixED, where sequences with an average length of $\leq 30$ tokens (see Fig. 4) perform significantly better than random: $\approx 7\%$ (vs. $2\%$ at random) for overall classification accuracy, $\approx 56\%$

[14]Available at https://github.com/AI-secure/aug-pe

(vs. $10\%$ at random) for category prediction accuracy, and $\approx 1.8$ (vs. $2.0$ at random) for average $L_1$ distance (lower is better) between predicted and true scores. We also note that in the case of open-ended text generation, such notions of downstream utility are not directly applicable. Thus, we adopt a combination of metrics to evaluate DP synthetic text.

### E.3 Top-$k$ Selection Statistics

Some additional statistics of selection from the expanded vocabulary described by the Top-$k+$ step of INVISIBLEINK are presented in Tab. 12. In particular, we show the following:

- $k_{\text{eff}}$: Effective-$k$, which is the size of the vocabulary $V_k^+$
- $\sigma_k$: standard deviation of $k_{\text{eff}} = |V_k^+|$
- #toks: number of tokens generated from the expanded vocabulary $V_k^+ \setminus V_k$ per generation

For each metric, we report the average over all generations or generated tokens for a setting. As the privacy budget increases, the clipping norm $C$ also increases. We observe that larger expansions of the top-$k$ threshold ($2C/B$), lead to larger effective-$k$ ($k_{\text{eff}} = |V_k^+|$) of Top-$k+$ sampling. Further, as reported in §5, we also observe that the number of tokens sampled from the expanded set is very low (typically $\leq 10$), supporting the claim that $V_k^+$ is a very tight superset of $V_k$. Further, the standard deviation of $k_{\text{eff}}$ scales roughly with $\sqrt{k_{\text{eff}} - k}$, i.e., the square root of the size of the expansion.

The number of sampled tokens also decreases as $k$ increases. Despite a larger increase in effective $k$ due to the expansion of the top-$k$ vocabulary, the probability of selection of tokens decreases sharply with rank since the increased expansion is offset by lower probabilities of selection.

| | | $k = 10$ | | | $k = 50$ | | | $k = 100$ | | | $k = 500$ | | |
|---|---|---|---|---|---|---|---|---|---|---|---|---|---|
| $\varepsilon$ | $C$ | $k_{\text{eff}}$ | $\sigma_k$ | #toks | $k_{\text{eff}}$ | $\sigma_k$ | #toks | $k_{\text{eff}}$ | $\sigma_k$ | #toks | $k_{\text{eff}}$ | $\sigma_k$ | #toks |
| **1.0** | 0.08 | 10.17 | 0.43 | 6.14 | 50.69 | 0.97 | 1.50 | 101.07 | 1.28 | 0.91 | 506.43 | 3.95 | 0.73 |
| **3.0** | 0.23 | 10.52 | 0.76 | 7.71 | 52.57 | 1.77 | 2.62 | 104.98 | 2.54 | 2.04 | 524.04 | 7.11 | 1.81 |
| **5.0** | 0.36 | 10.52 | 0.77 | 7.77 | 53.09 | 2.00 | 2.89 | 106.80 | 3.02 | 2.51 | 534.82 | 9.31 | 2.60 |
| **10.0** | 0.66 | 11.25 | 1.25 | 11.15 | 56.57 | 3.04 | 5.04 | 113.35 | 4.55 | 4.45 | 568.28 | 16.33 | 4.77 |

Table 12: Statistics for Top-$k+$ sampling step of INVISIBLEINK for 1000 synthetic text samples generated for the MIMIC dataset using a TinyLLaMA 1B model, batch size of $B = 7$, and sampling temperature $\tau = 1.2$. We report the size of $V_k^+$ ($k_{\text{eff}}$), the standard deviation in $k_{\text{eff}}$ ($\sigma_k$), and the number of tokens sampled from the expanded vocabulary $V_k^+ \setminus V_k$ (#toks); all metrics are reported averaged over all generations or generated tokens.

### E.4 Scaling INVISIBLEINK to Larger Models: Results on LLaMA3 8B

The utility scores (MAUVE) for 1000 synthetic texts generated using a LLaMA3 8B model are presented in Tab. 13. The results agree qualitatively with trends observed for smaller models: intermediate values of $k \approx 100$ to 1000 consistently perform better than very small or very large $k$.

| $\varepsilon$ | $k{=}10$ | $k{=}50$ | $k{=}100$ | $k{=}500$ | $k{=}1000$ | $k{=}5000$ |
|---|---|---|---|---|---|---|
| **1.0** | $58.9_{0.4}$ | $61.6_{0.5}$ | $\mathbf{62.9_{0.9}}$ | $62.4_{0.6}$ | $63.0_{0.0}$ | $61.8_{0.6}$ |
| **3.0** | $59.2_{0.6}$ | $60.0_{0.5}$ | $60.5_{1.8}$ | $62.1_{0.5}$ | $\mathbf{62.2_{0.5}}$ | $\mathbf{62.2_{0.5}}$ |
| **5.0** | $59.9_{0.6}$ | $60.7_{0.5}$ | $62.2_{1.1}$ | $62.8_{0.2}$ | $\mathbf{63.0_{0.3}}$ | $62.6_{0.3}$ |
| **10.0** | $59.4_{0.4}$ | $61.3_{0.8}$ | $62.9_{0.9}$ | $\mathbf{63.6_{0.3}}$ | $63.0_{0.0}$ | $63.2_{0.7}$ |

Table 13: MAUVE (%) scores, reported with mean and 95% confidence intervals (see §D.6) over 3 runs, for 1000 synthetic text generations using LLaMA3 8B for the MIMIC dataset with varying $\varepsilon$ and $k$ for INVISIBLEINK with $\tau = 1.2$ and $B = 7$. The best scores for every privacy budget are highlighted.

We note that generation from the LLaMA3 8B at such small batch sizes ($B = 7$) is *completely infeasible* for the other baselines we consider, such as AdaPMixED [19] and Amin et al. [13]'s method. As we use larger (in terms of number of parameters) LLMs, the computational cost in terms of both GPU memory and computational time increases sharply. Using batch sizes $> 50$, as proposed by the baselines [13, 19] is unreasonable with large models. This highlights the scalability of INVISIBLEINK to larger models, and by extension, its suitability for high-quality text generation.

However, we note that the overall MAUVE scores of the generated texts falls compared to generations from smaller models like TinyLLaMA and LLaMA3.2 1B. We attribute this to the strong alignment of the LLaMA3 8B model against generating (fake) medical records or divulging patient data. We further note that despite our detailed prompts (see Tab. 4), the output private text generated by the model contains text other than the synthetic text to be generated, usually in the form of disclaimers of the form: "Here is a fake patient discharge summary". Sanitizing the output private text, post-generation, by removing this disclaimer leads to $\approx 1 - 2\%$ increase in MAUVE scores across all settings.

While our primary experiments (reported in Fig. 4 and Tab. 2) show that INVISIBLEINK performs well, even when generating out-of-domain text, strong alignment of LLMs impacts generation adversely. We leave a full investigation of this phenomenon to future work.

### E.5 Ablation Studies: Effect of Temperature, Clip Norm, and Top-$k$ parameter

INVISIBLEINK has the following key hyperparameters: the batch size $B$ (which is determined by the compute budget), temperature $\tau$, clip norm $C$, and top-$k$ parameter. §5 shows the effect of varying the batch size $B$ (Fig. 4) the top-$k$ parameter (Fig. 6).

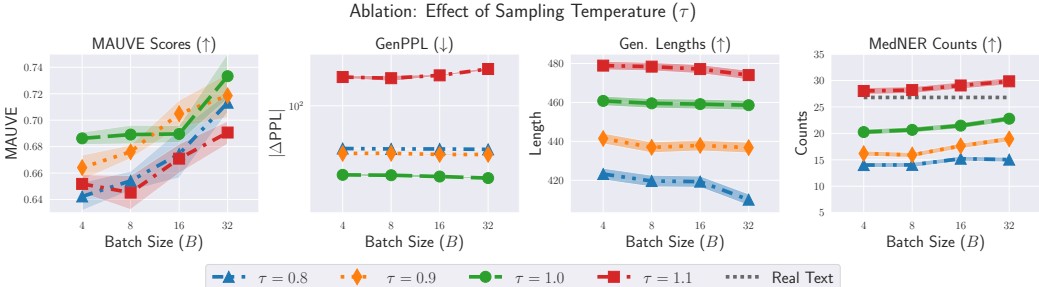

Figure 9: Variation of utility (measured using a variety of metrics) with sampling temperature, for the full-vocabulary variant of INVISIBLEINK ($k = |V|$), reported for 1000 synthetic generations, averaged over 3 runs, for the MIMIC dataset using a TinyLLaMA 1B model. Temperature is varied from 0.8 to 1.1 for a fixed privacy budget $\varepsilon = 5$. We observe that selecting a temperature of $\tau \approx 1.0 - 1.1$ consistently gives better performance. When coupled with Top-$k$+ sampling for $k \approx 50 - 100$ (chosen based on empirical observations in Fig. 2), we observe that the best performance is obtained for $\tau = 1.1$. We report these results in the main paper; in Fig. 4.

As per the heuristics for hyperparameter selection proposed in §5, we tune across various temperatures and calculate the clip norm $C$ for the fixed privacy budget and sampling temperature $\tau$. The effects of varying the temperature and clip norm are thus coupled. We first report the performance of INVISIBLEINK for various temperatures and top-$k$ parameters. We observe from Fig. 9 & Fig. 10 that the optimal choice of sampling temperature and top-$k$ parameter is $\tau = 1.1$ and $k = 100$.

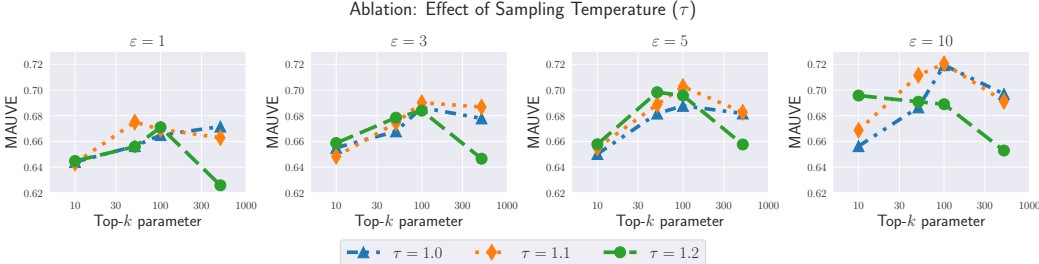

Figure 10: Variation of utility (measured using MAUVE scores) with sampling temperature and top-$k$ parameter, reported for 1000 synthetic generations, over one run, for the MIMIC dataset using a TinyLLaMA 1B model. Temperature is varied from 1.0 to 1.2 for various privacy budgets. We observe that selecting a temperature of $\tau \approx 1.1$ consistently gives the best performance when coupled with Top-$k$+ sampling for $k \approx 50 - 100$ (chosen based on empirical observations in Fig. 2). We report these results in the main paper; in Fig. 4.

We now decouple the effect of clipping by removing the constraint on privacy budget and instead varying $C$ freely for a given fixed temperature (see Fig. 11). We also report the variation of calculated clip norm for INVISIBLEINK and Amin et al. [13]'s method with batch size $B$ for various privacy

budgets. As discussed in Fig. 3, the clipping norms required for high-utility generation using INVISIBLEINK can be achieved with much lower batch sizes as compared to Amin et al. [13].

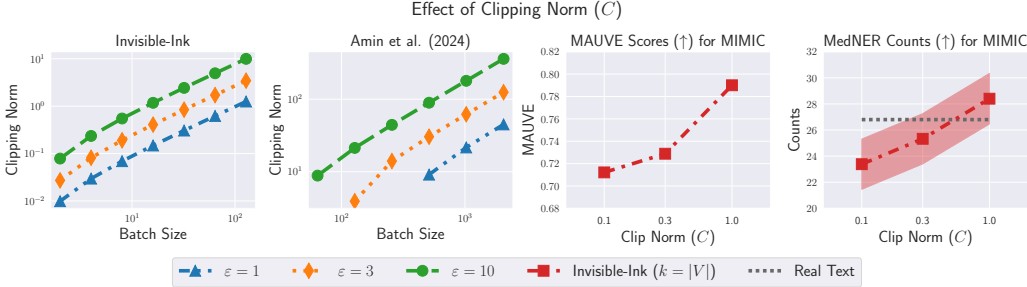

Figure 11: Effect of clipping norm. **Left two:** Variation of calculated $C$ for INVISIBLEINK and Amin et al. [13]'s method with batch size $B$ for various privacy budgets. The latter needs much larger batch sizes to give comparable clip norms for a given privacy budget. **Right two:** Variation of utility with $C$ for 1000 synthetic generations for the MIMIC dataset using a TinyLLaMA 1B model using INVISIBLEINK ($k = |V|$). Temperature ($\tau = 1.2$) and batch size ($B = 7$) are fixed while we vary the clip norm. Each setting of the clip norm implies a DP parameter as given by Theorem 2. In particular, it varies from $\varepsilon = 1.3$ for $C = 0.1$ to $\varepsilon = 17.6$ for $C = 1.0$.

## E.6 Comparison of Adjacency Notions for AdaPMixED

As discussed in §B.3, we use an implementation of AdaPMixED that employs the *replace-by-null* notion of adjacency, as in INVISIBLEINK, in the main paper instead of the *add-or-remove* notion of adjacency used by Flemings et al. [19] as it allows for an apples-to-apples comparison.

| $\varepsilon$ | Add-or-remove adjacency | | | Replace-by-null adjacency | | |
|---|---|---|---|---|---|---|
| | $B = 8$ | $B = 32$ | $B = 128$ | $B = 8$ | $B = 32$ | $B = 128$ |
| 1.0 | 58.68 | 58.71 | 58.51 | 58.14 | 57.64 | 59.51 |
| 3.0 | 58.29 | 58.81 | 61.50 | 58.30 | 57.97 | 58.80 |
| 5.0 | 60.24 | 58.72 | **TLE** | 58.26 | 59.32 | **TLE** |
| 10.0 | 60.23 | 59.23 | **TLE** | 59.18 | 59.61 | **TLE** |

Table 14: MAUVE scores (%) for 1000 synthetic generations (over one run only) for the MIMIC dataset using TinyLLaMA for different privacy budgets ($\varepsilon$) and batch sizes ($B$), comparing two adjacency notions: **Add-or-remove** and **Replace-by-null** for AdaPMixED. **TLE** denotes runs where a wall-clock time limit of 24 hrs was exceeded. Top-performing configurations per $\varepsilon$ tend to vary with adjacency notion and batch size; highlighting the sensitivity of MAUVE scores to both.

To do so, we modify the data-dependent privacy accounting step (defined in [19, Eqn. 3]), which accounts for the maximum symmetric $\alpha$-Rényi Divergence between probability distributions corresponding to the reference set and all its possible neighbours under *add-or-remove* adjacency.

We modify this step to instead account for the maximum $\alpha$-Rényi Divergence between the probability distribution corresponding to the current reference set and all possible neighbours using the *replace-by-null* adjacency notion. In contrast to the *add-or-remove* notion, where removing a particular reference zeroed out its contributing probabilities [19, Alg 1, Line 15], in our modified algorithm, replacing a particular reference by "null" (empty string $\emptyset$; see Theorem 5) means that the contributing probabilities default to that of the public distribution. Mathematically, we modify [19, Alg 1, Line 15] to be (borrowing all notation from [19]):

$$p_{-i}(\boldsymbol{x}) = \frac{1}{N} \left( \sum_{j \neq i} \bar{p}_j(\boldsymbol{x}) + p_{\mathsf{pub}}(\boldsymbol{x}) \right)$$

We note that this lends an advantage to the *replace-by-null* implementation of AdaPMixED, since the per-token privacy accounted for must be slightly lesser. However, the quantitative results, presented in Tab. 14, fail to reflect this meaningfully, since the generated sequences are very small. Regardless, INVISIBLEINK outperforms AdaPMixED significantly irrespective of the notion of adjacency used.

# F  Limitations, Discussion, and Broader Impact

INVISIBLEINK casts next-token sampling as an instance of the exponential mechanism over LLM logits and introduces two key innovations. First, it reduces privacy cost by isolating and clipping only the sensitive portion of the logits, measured relative to public logits. Second, it improves utility by sampling from a tight superset of the top-$k$ private tokens. Empirical results demonstrate an $8\times$ (or more) reduction in compute cost compared to state-of-the-art baselines, while preserving utility, across privacy levels. Notably, INVISIBLEINK enables private long-form generation at less than $4\text{-}8\times$ the cost of non-private decoding, offering a practical and efficient solution for privacy-preserving language generation. While INVISIBLEINK significantly reduces the computational complexity of generating DP text, it still has some limitations that we now discuss.

**Limitations.** The trust model of INVISIBLEINK requires specific assumptions that could potentially be restrictive. A direct implementation of INVISIBLEINK requires a trusted central aggregator who has access to the raw sensitive references, whitebox access to the model weights (to query the next-token logits), and the necessary computational resources to run INVISIBLEINK. Relaxing these requirements is an interesting avenue for future work.

Second, INVISIBLEINK provably protects against privacy leakage through inference, but not in the training or fine-tuning stage. Leakage could still be possible through data contamination or backdoor adversarial attacks. Furthermore, even in the case of inference, we tackle example-level DP only. However, user-level privacy leakage is also possible [e.g. 96], and extending INVISIBLEINK to user-level DP is an interesting direction.

Finally, while INVISIBLEINK drastically improves the computational cost of private text generation, the $8\times$ overhead over non-private generation could still be restrictive with large frontier models under extremely tight privacy budgets. Pushing the envelope of the privacy-utility-compute tradeoffs of text generation is a fruitful avenue for future research.

**Broader Impact.** The development of computationally efficient DP text generation can allow safe deployment of large language models (LLMs) in privacy-sensitive domains such as healthcare, law, and finance. By addressing the steep privacy-utility-compute tradeoffs that currently hinder practical use, our work could help unlock high-impact applications that require both strong privacy guarantees and high-quality long-form generation.

In doing so, this research contributes to sustaining public trust in AI systems. As LLMs become increasingly integrated into everyday decision-making processes, ensuring that user data is handled with dignity and protected by rigorous guarantees is ethically and, in sensitive domains like healthcare, legally warranted. We view computationally efficient and provably privacy-preserving techniques as key to building ethical and trustworthy AI.

However, the existence of formal privacy guarantees may lead to overconfidence or misuse, particularly if system integrators overlook practical limitations such as implementation errors, misconfigured parameters, or unrealistic threat models [e.g., 82, 83]. Differential privacy is a powerful but narrow tool: it provides provable anonymization but is only one component in an end-to-end system. The practical use of such a system must be considered in a holistic manner, along with clear communication of failure modes.

# G  Software Package

We illustrate the use of the accompanying software package, available on GitHub,[15] and installable via pip,[16] as `pip install invink`.

```
1 import invink
2 from datasets import load_dataset
3
4 # Load the TAB dataset for this example.
5 data = load_dataset("mattmdjaga/text-anonymization-benchmark-train")
6
```

---

[15]https://github.com/cerai-iitm/invisibleink
[16]https://pypi.org/project/invink/

```
7  # Reference texts given as a list of strings.
8  ref_txts = data["train"]["text"]
9
10 # Use smaller models for lower memory footprint.
11 model = "google/gemma-3-4b-it"
12
13 # Description should not contain privacy-sensitive information.
14 dataset_desc = "The dataset comprises English-language court cases
       from the European Court of Human Rights (ECHR)."
15
16 # Generate 10 text samples with (10, 1e-5)-DP
17 output = invink.generate(ref_txts, model, num=10, epsilon=10,
18     batch_size=8, max_toks=500, dataset_desc=dataset_desc)
19
20 # list of private synthetic text strings
21 print(output.texts)
```

Listing 1: Generate private synthetic text using INVISIBLEINK

The demo code in Listing 1 reads reference texts from the TAB dataset and generates num=10 synthetic texts, each at most $500$ tokens long, with a $(\varepsilon = 10, \delta = 10^{-5})$-DP guarantee. The parameter batch_size corresponds to $B + 1$ in the paper, i.e., the total number of LLM inferences per generated token. Thus, $(\texttt{batch\_size}-1) \times \texttt{num} = (8-1) \times 10 = 70$ private references are used to generate 10 synthetic texts in the above example.

## H   Example Generations

See Tab. 15 for some example generations from INVISIBLEINK.

| Dataset | Examples of Generated Synthetic Text |
|---------|--------------------------------------|
| Yelp | "We chose a cozy restaurant that delivered impeccable service, excellent fare, and even better music to our table." (ReviewStars: 4.8 out of 5.0) Naturally, we walked in to find a spacious and serene lobby. Our server was welcoming, and she immediately recommended a seasonal cocktail and the restaurant's signature dish, Chicken Caesar. The servers were friendly, knowledgeable, and efficient which contributed greatly to our experience. Tucking into each other's orders, we feasted our eyes on sophisticated appetizers like Caprese, Grilled Salmon, and Grilled Caesar Wings, accompanied by the juiciest and most lip-smacking chicken you'll ever taste, alongside a delicate glass of Pinot Noir to top it off. The food arrived quickly without forgetting the music |
|  | "Although the staff may not be able to befriend every customer, their attentiveness and positive attitude really go a long way. Our recent visit to a new coffee shop led by a chatbot could have gone further in helping patrons' preferences, with many potential avenues to cater the atmosphere. From a personal observation, however, I believe that adding more personalities to the virtual assistants could cater to a wider-range of needs while engaging customers. Nevertheless, the chatbot is smart and proficient in its tasks, and we enjoyed getting tips about the local coffee culture from their knowledgeable baristas. Although we were a bit inconveniently served at our preferred location for a quick lunchtime fix, we were glad to taste some delicious homemade sushi on the go. Overall, our overall experience was positive with a bit more effort and personalization in terms of addressing specific preferences could lead to repeat vis |
| TAB | PROCEDŽURE: The case originated in an Application for Judicial Review filed by Ms. Smith, petitioner(s), against the Bulgarian State from January 8th of the year 2019. The Respondent for this jurisdictional issue is the Ministry of the Interior of the Bulgarian Republic and the Bulgarian National Antiterrorism Bureau. The alleged event occurred at the intersection of a public place known as Sidi Boutha Crescent or Vezhub Khayri Crescent in the city of Sofia, Bulgaria that caused or contributed significantly and directly to plaintiff Smith's physical injury causing death pursuant to Section 4, Article 3 of the Bulgarian Medical Registration Act.
- The procedural steps involved in the application included the preparing of the written petition as well as its formality submitted to the Court for review.
- Upon review of the documents, the Registrar of the Court sent the petition to the Bulgarian Ministry of the Interior as a legal source and required from the said organization to verify and respond in court about the availability of sufficient grounds for such judicial review against the Bulgarian state.
- Following the review process, the request for verification made by plaintiff Smith's attorney was submitted before the Ministry of the Interior of the Bulgarian Republic.
- The Bulgarian Minister of the Interior received the request as per the requirement and made a decision within one month from the submission date based on the relevant document provisions as provided by law.
- After this, the Ministry communicated to the plaintiff regarding the reply received from the National Antiterrorism Bureau, and the said communication stated that the respondent is satisfied with having received the reply and that the Bulgarian state is willing to conduct a judicial investigation of the event from the incident that caused or contributed significantly to the death of the petitioning defendant Smith.
Respectfully, as per the international human rights instrument cited in the previous reference to the jurisdiction as mentioned in the jurisdictional challenge brought in favor of Ms. Smith, all facts and circumstances of the criminal matter that led to the loss of a life of a Bulgarian citizen constitutes a violation of her right to physical integrity protected by Article 2 (Prohibition of T |

Table 15: Randomly chosen examples of private text generated with $\varepsilon = 10$ from the Yelp and TAB datasets using the TinyLLaMA 1B model, sampling temperature $\tau = 1.2$, $k = 50$, and batch sizes $B = 31$ and $B = 7$ respectively; INVISIBLEINK provides high-quality text. We do not provide examples of synthetic generations for the MIMIC dataset to comply with its license. Amin et al. [13]'s method is infeasible for text generation at the same batch sizes.

