# OpenReview forum: "InvisibleInk: High-Utility and Low-Cost Text Generation with Differential Privacy"
_NeurIPS.cc/2025/Conference — NeurIPS 2025 poster_

### Official Review · Reviewer_wRyU · 2025-06-30

**Clarity:** 2
**Significance:** 3
**Originality:** 2
**Rating:** 4
**Confidence:** 4

**Summary:**

This paper proposes an efficient differential privacy long text generation framework called InvisibleInk, which aims to solve the problem of protecting sensitive information when generating text while ensuring the quality of the generated text and low computational cost.

**Questions:**

For specific questions, please see "Weaknesses".

**Ethical Concerns:**

["NO or VERY MINOR ethics concerns only"]

**Final Justification:**

I have read the authors' response. Thanks for the rebuttal. The response mainly addressed my concerns. I will raise my rating accordingly.

**Limitations:**

Yes

**Quality:**

2

**Strengths And Weaknesses:**

Strengths：

1.The two core technologies proposed in this paper, DClip and Top-k+ sampling, fundamentally solve the pain points of high computational overhead and poor text quality in traditional DP text generation. DClip avoids redundant privacy consumption of language prior knowledge by directional clipping of sensitive information differences, while Top-k+ sampling cleverly combines the efficiency of non-private decoding with the privacy protection of DP.

2.By deeply combining differential privacy with the LLM generation process, a complete long text generation framework was constructed. Different from the previous method of simply applying the exponential mechanism, the design of InvisibleInk is more in line with the generation characteristics of LLM, providing new ideas for the application of DP in the field of NLP.

3.Experiments are conducted on datasets from three different fields: medical, legal, and commercial, including MIMIC-IV, TAB, and Yelp, to verify the universality and cross-domain adaptability of the framework.

Weaknesses：

1. The privacy protection capability of the algorithm has not been verified under specific attack scenarios (e.g. membership inference attack).

2. The algorithm description in Figure 1 is too simple, and the algorithm execution process cannot be intuitively seen, especially Top-k+ sampling. I don’t know what the green dotted line means.

3. The horizontal axis of Figure 3 is not clearly stated.

4. Why is privacy loss of +∞ in line 190?

5. Why is Top-k+ sampling constructed? The full text does not seem to explain it clearly, so I don’t know whether it is necessary.

6. This method is only applicable to white-box LLMs, and cannot benefit from the closed-source black-box LLMs.

7. The experiment only compares the MAUVE scores of synthetic text and reference text, which can only reflect the similarity of the overall distribution, and does not test the utility on specific downstream tasks. The RAG mentioned in the abstract and introduction does not seem to have specific experiments in this scenario.

---

> ### Author Rebuttal · Authors · 2025-07-31
>
> We thank the reviewer for the thorough review and the questions they have raised. We answer each one in detail.
>
> > The privacy protection capability of the algorithm has not been verified ...
>
> Following established practice in the literature, we do not perform attack evaluations. Most prior works in private text generation [1-6] do not conduct additional study on the effectiveness of privacy protection when subjected to membership inference attacks. This is similar in spirit to DP model training, where one does not usually evaluate the effectiveness of training algorithms with membership inference attacks.
>
> As an aside, we note that research on designing effective membership inference attacks for text generated by LLMs is a new and emerging research topic ( the first promising work in this field is the recent paper by Meeus et al. (ICML 2025) [7]).
>
> > … description in Figure 1 is too simple ... what the green dotted line means.
>
> The green dotted line in Figure 1 is the top-k threshold of the public logits. Only tokens in the clipped-and-aggregated logits that lie in the public top-k set are selected in naive top-k sampling; the rest are discarded.
>
> However, when private logits are clipped and aggregated using DClip, individual private logits may cause a deviation of at most C/B to the public logit values of every token. In particular, some tokens outside the public top-k set may now clear the threshold. Top-k+ sampling (modified threshold denoted by the solid green line) allows us to recover a few such tokens, which would otherwise have been discarded with naive top-k sampling.
>
> We would be happy to elaborate on any specifics and update the figure as required.
>
> > The horizontal axis of Figure 3 is not clearly stated.
>
> - Leftmost fig.:  The horizontal axis denotes the value of the logits $\phi_i(y)$ for different values of the index $i$ and the token $y$. This is calculated based on one batch of size B=127 from the MIMIC data, pooled over 500 generated tokens.
> - Middle left fig.: Same setting as the above, but the horizontal axis now denotes the difference of private and public logits, i.e., $\phi_i(y)  - \phi_{pub}(y)$ for different values of the index $i$ and token $y$.
> - Right two fig.: The horizontal axis denotes the token index, when ranked in decreasing order of the unclipped logit value (which we refer to as token rank). We select an unclipped private logit vector from the MIMIC data generation and compare the distortion introduced by using the clipping function of Amin et al. (2024) [2] and our proposed clipping function, DClip.
>
> We will make this clear in the caption.
>
> > Why is privacy loss of +∞ in line 190?
>
> When the inclusion or exclusion of a sensitive reference causes a token to "jump out" of the top-k set, its probability of being generated goes from a non-zero value to a zero value. In such a case, the privacy loss random variable takes a value of $+\infty$. We elaborate on this below.
>
> We first recall the definition of the privacy loss (e.g. Definition 2 of Steinke (2022) [8]).  Let $P$ and $Q$ be 2 probability distributions on $\mathcal{Y}$, which correspond to the outputs of a randomized mechanism on two “neighbouring” inputs. The privacy loss random variable, denoted by $Z$, is defined as $Z=f_{P||Q}(Y)=log(P(Y))/log(Q(Y))$ where $Y$ is distributed according to $P$.
>
> Suppose a given token $y$ lies in $\bar{V}_k$ as defined in line 183 of the main paper for a given set of references $R=(r_1, \dots, r_B)$. Let $y\notin \bar{V}_k$ for some $R’$ adjacent to $R$. Note that this is possible since $\bar{V}_k$ is dependent on the clipped private logits $\phi^{\text{clip}}_i$. Since we discard all tokens not in $\bar{V}_k$, the probability of token $y$ being output for the reference set $R$ and $R’$ are non-zero (i.e. $P(y) \neq 0$) and zero (i.e. $Q(y) = 0$), respectively. Thus, the corresponding value of the privacy loss random variable at $Y=y$ is $\log( P(y) / Q(y)) = +\infty$.
>
> Note that when the set of discarded tokens does not depend on the private logits, moving from reference set $R$ to $R’$ does not affect the probabilities of outputting any given token. Thus, privacy loss remains zero for all tokens, i.e., no privacy is leaked.
>
> > Why is Top-k+ sampling constructed ...
>
> Naively sampling the top-k tokens from the public logits potentially discards tokens that are highly ranked in the private logits. Such tokens are likely to be relevant to the data generation domain, and it is thus desirable to recover them. However, selecting them based on the private logits leaks privacy (as above).
>
> To circumvent this issue, we design Top-k+ sampling. We explain this step-by-step below:
>
> - We calculate the top-k threshold $l$ of the public logit $\phi_{\text{pub}}$
> - We lower the threshold by $2C/B$. The new threshold $l’ = l - 2C/B$ (Note that this modified threshold is still agnostic to private logits)
> - We select all tokens in $V$ that have public logit values higher than $l’$ into $V_k^+$.
>
> To summarize, we construct a tight superset of the public top-k token set by lowering the public top-k threshold by $2C/B$.
>
> When private logits are clipped and aggregated using DClip, individual private logits may cause a deviation of at most $C/B$ to the public logit values of every token. Let $\phi_i^{\text{clip}} = \phi_{\text{pub}} + \frac{1}{B}\text{clip}\_C(\phi_i, \phi_{\text{pub}})$ be the contribution of one private logit. Then for a token $y\in V$ (token indexed by $k$, i.e. $\phi(k)$ refers to top-k$^{th}$ token):
>
> $$ \phi_i^{\text{clip}}(y) \geq \phi_i^{\text{clip}}(k) \Rightarrow \phi_{\text{pub}}(y) + \frac{1}{B}\text{clip}\_C(\phi_i(y), \phi_{\text{pub}}(y)) \geq \phi_{\text{pub}}(k) + \frac{1}{B}\text{clip}\_C(\phi_i(k), \phi_{\text{pub}}(k)) \Rightarrow \phi_{\text{pub}}(y) \geq \phi_{\text{pub}}(k) - 2C/B,$$
>
> where the last step uses the fact that each $\text{clip}\_C(x,y)$ is bounded to lie in the range $[-C, +C]$. Since $C\ll B$ typically, and $|V_k^+ - V_k(\phi_{\text{pub}})| \ll |V|$ from Table 11 (p.39 App. E2), we see that $V_k^+$ is indeed a tight superset of the public top-k token set.
>
> We would be happy to add this discussion to the revised draft.
>
> > This method is only applicable to white-box LLMs...
>
> While LLMs like Claude, GPT4, and Gemini are hidden behind an API, there is an increasing number of state-of-the-art general-purpose and reasoning open-weights models like DeepSeek, LLaMA3, and Mistral. Such models allow white-box logit access and are compatible with InvisibleInk.
>
> As demonstrated in Fig. 6 (p. 38, App. E) of the supplement, leveraging the whitebox access to the logits allows us to significantly outperform the state-of-the-art API-access method AugPE [9].
>
> Further, white-box access to next-token logits is a commonly studied setting in the literature on DP text generation. For example, see [1-4]. InvisibleInk operates under the same assumptions but demonstrates an order-of-magnitude improvement in the privacy-utility-compute tradeoff.
>
> > ... does not test the utility on specific downstream tasks ...
>
> Following the reviewer’s excellent suggestion, we ran additional experiments to evaluate the synthetic data on a downstream task. We finetune a RoBERTa model for sequence classification of the generated synthetic text from the Yelp dataset into 50 classes, obtained as all combinations of 5 review score labels and 10 business category labels. (Since the synthetic data generation is conditioned on these axes, the class label information is also available in the synthetic data.)
>
> We report the following metrics:
>
> - Classification Accuracy: correct predicted class (%age)
> - Top-5 Accuracy: correct class in the top-5 predictions (%age)
> - Category Accuracy: correct business category (%age)
> - Score Accuracy: correct review score (%age)
> - Score L1: Average L1 distance between predicted and true review scores
>
> The results presented in the table compare the downstream utility metrics for text generated by various methods with a fixed privacy budget of $\varepsilon=10$.
>
> | Method | Accuracy ($\uparrow$) | Top-5 Acc. ($\uparrow$) | Cat. Acc. ($\uparrow$) | Score Acc. ($\uparrow$) | Score L1 ($\downarrow$) |
> |---|---|---|---|---|---|
> | InvisibleInk (B=8) | **32.98**  | **72.16**  | **64.90** | **52.2**  | **0.652** |
> | Amin et al. (2024) (B=128) | 29.44 | 64.56 | 60.18 | 46.64  | 0.748 |
> | AdaPMixED (B=32)  | 7.44 | 34.72 | 56.82 | 15.44  | 1.858 |
>
> InvisibleInk outperforms all baseline methods by a large margin at a significantly smaller computation cost, even when measured in terms of this downstream task utility. We will include these in the revision. These results are similar to MAUVE and overall text similarity metrics.
>
> Apart from MAUVE scores, we also report additional evaluation metrics, such as MedNER counts, average generation lengths, and the difference in generation perplexity between real and synthetic data; see App. D4 (pp. 33-34).
>
> [1] Amin, et al. "Private prediction for large-scale synthetic text generation." EMNLP (2024).
>
> [2] Flemings et al.. "Differentially private next-token prediction of large language models." NAACL-HLT 2024.
>
> [3] Flemings et al.. "Adaptively private next-token prediction of large language models." (2024).
>
> [4] Tang et al. "Privacy-preserving in-context learning with differentially private few-shot generation." ICLR (2024).
>
> [5] Yue, et al. "Synthetic text generation with differential privacy: A simple and practical recipe." ACL 2023.
>
> [6] Yu, et al. "Privacy-preserving instructions for aligning large language models." ICML 2024.
>
> [7] Meeus, et al. "The Canary's Echo: Auditing Privacy Risks of LLM-Generated Synthetic Text." ICML (2025).
>
> [8] Steinke. "Composition of differential privacy & privacy amplification by subsampling." (2022). Chapter 3 of the book “Differential Privacy in Artificial Intelligence: From Theory to Practice.”
>
> [9] Xie, et al. "Differentially Private Synthetic Data via Foundation Model APIs 2: Text." ICML 2024.

---

> > ### Author Response · Authors · 2025-08-07
> > **Thank you for your review! Could you please check the rebuttal?**
> >
> > Thank you again for the review. As the discussion period draws to a close, we request you to take a moment to consider our response.
> >
> > We would be happy to elaborate on or clarify any points as necessary. If our responses have addressed your concerns, we would be grateful if you would consider updating your assessment accordingly. Thank you very much!

---

> > > ### Author Response · Authors · 2025-08-09
> > > **Thank you again! Could you please check our rebuttal?**
> > >
> > > We thank the reviewer again for the constructive review.
> > >
> > > As the discussion period draws to a close, we would like to gently request you to consider updating your assessment, based on the clarifications and further experimental results we have presented during the rebuttal phase.
> > >
> > > If there are any further questions or concerns, please do not hesitate to let us know, and we will do our best to answer them in the time that remains.
> > >
> > > Thank you very much!

---

### Official Review · Reviewer_8mTj · 2025-06-30

**Clarity:** 3
**Significance:** 2
**Originality:** 2
**Rating:** 4
**Confidence:** 4

**Summary:**

This paper proposes *InvisibleInk*, a novel framework for differentially private (DP) text generation that targets both high utility and computational efficiency. The core idea is to aggregate token-level logits from private reference data using a new *DClip* mechanism, which clips their differences against a public logit vector. To reduce computational cost, the authors further introduce a *Top-k⁺* pruning technique that limits the sampling space without additional privacy expenditure. The method is analyzed under the zCDP framework, and experiments demonstrate strong performance relative to prior work, with better trade-offs between privacy and utility.

**Questions:**

See weaknesses.

**Ethical Concerns:**

["NO or VERY MINOR ethics concerns only"]

**Final Justification:**

The rebuttal has resolved most of my concerns. I have updated my score to 4 accordingly.

**Quality:**

2

**Strengths And Weaknesses:**

**Strengths:**
1. The paper tackles an practical challenge: how to conduct efficient and private text generation. The proposed method is compatible with autoregressive language models and can be deployed in real-world settings where public references are available.
2. The modular design of *DClip* and *Top-k⁺* filtering makes the method potentially applicable to a wide range of DP text generation pipelines.

**Weaknesses:**
1. The assumption of “zero-cost” privacy for public information in DClip is overly idealized. A central innovation of *InvisibleInk* is the construction of aggregated logits via clipping against public logits ($ϕ_{pub}$), which assumes $ϕ_{pub}$ is non-sensitive. However, since $ϕ_{pub}$ is computed on "*prompt + prefix $x_{<t}$*", the prompt may still reveal private user intent or context, and therefore the assumption of public safety may not always hold.
2. The privacy composition analysis assumes each token-generation step is independent and uses standard composition under zCDP. However, text generation is inherently *contextual and autoregressive*, meaning that each token depends on previously generated tokens. This violates the independence assumption, especially for long-form generation, and could lead to an underestimation of true privacy leakage.
3. While *InvisibleInk* performs well in white-box settings, most widely deployed LLMs (e.g., GPT-4, Claude, Gemini) are only accessible via black-box APIs. The proposed method cannot be directly applied to such settings, limiting its practicality.
4. The experimental scope is relatively narrow, using only three datasets (MIMIC-IV, Yelp, and TAB). For stronger empirical support, additional datasets should be included, such as AGNews, TREC, DBPedia, IMDB, WikiMovies, and JSON, to improve the comprehensiveness and credibility of the evaluation.
5. The paper lacks analysis of generalization and robustness with respect to key hyperparameters (e.g., τ, C, k). Understanding the sensitivity of the method to these parameters is crucial for assessing its stability and deployment readiness.

---

> ### Author Rebuttal · Authors · 2025-07-31
>
> We thank the reviewer for the review and answer their questions below.
>
> > The privacy composition analysis assumes ...
>
> The adaptive composition property of DP allows us to analyze privacy in settings such as autoregressive text generation, where each token is generated based on the previously generated tokens. Below, we recall the adaptive composition property of zero-concentrated DP and instantiate it more precisely in our setting. For precise reference, see Proposition 19, [1], also recalled in Lemma 13 (pp. 29-30, Appendix C).
>
> **Lemma (Adaptive Composition of zCDP)**:
> If a mechanism $\mathcal{A}_1: \mathcal{X}^* \rightarrow \mathcal{Y}_1$ is $\rho_1$-zCDP and a mechanism $\mathcal{A}_2: \mathcal{X}^* \times \mathcal{Y}_1 \rightarrow \mathcal{Y}_2$ is $\rho_2$-zCDP, then their (adaptive) composition $\mathcal{A}: \mathcal{X}^*\rightarrow \mathcal{Y}_2$ defined by $\mathcal{A}(x) := \mathcal{A}_2(x, \mathcal{A}_1(x))$ is $(\rho_1+\rho_2)$-zCDP.
>
> For simplicity, consider the setting of generating two tokens in an autoregressive fashion. (The same can be extended to multiple tokens by induction.)
>
> Here, each $x \in \mathcal{X}^*$ denotes a set of sensitive input references. $\mathcal{Y}_1$ and $\mathcal{Y}_2$ denote the set of all possible output sequences of length 1 and 2, respectively, ie, $\mathcal{Y}_1 = V^1$ and $\mathcal{Y}_2 = V^2$, where $V$ is the model vocabulary. $\mathcal{A}_1$ denotes the generation of the first token given the input, while $\mathcal{A}_2$ denotes the generation of the second token given the input prompt and the first token. We see that $\mathcal{A}$ generates the two-token sequence from the input alone, with an overall $(\rho_1+\rho_2)$-zCDP privacy guarantee, where the second token generation is conditioned on the output of the first token.
>
> Thus, the dependence of a given token on previously generated tokens is handled gracefully by adaptive composition, and the privacy loss incurred is also correctly accounted for in the analysis. We are happy to elaborate on this in the next revision of the paper.
>
> Finally, we note that other works in the text generation setting also rely on the same adaptive composition property, e.g. [2].
>
> > While InvisibleInk performs well in white-box settings ...
>
> While LLMs like Claude, GPT4, and Gemini are hidden behind an API, there is an increasing number of state-of-the-art general-purpose and reasoning open-weights models like DeepSeek, LLaMA3, and Mistral series of models. Such models allow white-box access to next-token logits and are compatible with InvisibleInk.
>
> Further, white-box access to next-token logits is a commonly studied setting in the literature on DP text generation. For example, see [2, 4, 5, 6]. InvisibleInk operates under the same assumptions but demonstrates an order-of-magnitude improvement in the privacy-utility-compute tradeoff even over previous SoTA *whitebox* methods.
>
> > The experimental scope is relatively narrow ...
>
> We respectfully disagree with the reviewer’s assessment that the experimental scope is relatively narrow for the following reasons:
> - Three diverse and carefully chosen domains. In particular, the MIMIC (real medical discharge summaries) and TAB (real ECHR case documents) contain real privacy-sensitive data, making the evaluations more realistic than treating non-private internet data (that the models have likely already seen in pretraining) as “private” for the sake of evaluations. We elaborate on this further below.
> - We also respectfully point out that several previous works in this field have demonstrated the efficiency of their approaches on 2-4 datasets, for example: one [9], two [4,7,8], or four [5] datasets.
> - We give an exhaustive comparison to various state-of-the-art baselines from diverse families of algorithms, including a previous exponential mechanism-based generation approach (Amin et al. (2024) [2]), a private next-token prediction approach (AdaPMixED [5]), and a blackbox evolutionary algorithm (AugPE [3]).
> - We perform these evaluations across model scales from 1B to 8B parameters.
> - We conduct detailed analyses on the effect of all important hyperparameters (see also response to your question on this).
> - We also give practical recommendations and heuristics (p.9) on how to select the hyperparameters to use InvisibleInk effectively in new settings with minimal tuning.
> - The unanimous conclusion across all of these experiments is that InvisibleInk enjoys a *consistent 8x improvement* (or better) on the computational cost of generating high-quality synthetic data at the same privacy level.
>
> Thus, we argue that our experimental evaluation is, in fact, extensive and comprehensive (in the sense that all relevant scientific questions have been addressed thoroughly in diverse settings). If the reviewer can point out precise scientific questions that need to be addressed with additional benchmarks/experiments, we would be happy to consider them.
>
> **Why not other datasets?** We did consider the various datasets pointed out by the reviewer. Yet, we preferred our datasets for the following reasons.
>
> First, we specifically target  *long-form* text generation (e.g. >= 500 tokens) in this paper. This rules out *short-form* question answering and classification datasets such as AGNews, TREC, DBPedia, and WikiMovies as they do not contain sufficiently long real text for generation (<200 tokens on average).
>
> Further, these datasets, and others such as IMDB, are well-represented in the pretraining data of most LLMs. It is well-established in the literature that a high degree of overlap between the pretraining and target data distributions skews the conclusions of private data analysis [10]. In particular, publicly prompted LLMs can generate high-quality synthetic text for such datasets. Thus, researchers and thought leaders advocate the use of benchmarks that are sufficiently out-of-distribution relative to the training data and preferably contain real privacy-sensitive data, rather than treating non-privacy-sensitive internet data as “private” for experimentation [10].
>
> Accordingly, we select the MIMIC dataset to be our primary dataset for experiments. It is a dataset of real patient discharge notes containing sensitive medical data, and cannot legally be used for pretraining LLMs as per its data use agreement. Similarly, the TAB dataset contains sensitive identifying information (e.g. demographic traits, spatio-temporal markers, biographical details) from European Court of Human Rights cases. We use the Yelp dataset as a representative dataset that overlaps significantly with the pretraining data (similar to the other datasets proposed by the reviewer).
>
> > ... lacks analysis of generalization and robustness with respect to key hyperparameters …
>
> We have already performed all these sensitivity and robustness analyses. We refer to the following sections in the main paper and the supplement:
>
> - Effect of temperature $\tau$: Figure 7 (p. 40, Appendix E) shows the corresponding effect of the temperature on InvisibleInk. These results are similar in spirit to those in Figure 2 of the main paper, which shows the effect of the temperature in the *non-private* setting.
>
> - Top-k parameter: Figure 5 (p. 9, Section 5) illustrates the effect of the top-k parameter. We observe that intermediate values of k=50 or 100 show the best performance. This is further detailed in Table 11 (p. 39, App. E) and Table 12 (p. 40, App. E) in the supplement.
>
> - Clip norm $C$; Figure 8 (p. 41, Appendix E) explores the effect of the clip norm parameter.
>
> In short, the dependence on these hyperparameters is intuitive and predictable. We recommend practical heuristics for selecting a good set of hyperparameters with minimal tuning cost in the discussion on “Practical Recommendations” in Page 9 of the main paper. We would be happy to elaborate further on these points if necessary.
>
> We would also be happy to consider other suggestions on ablations or additional analyses.
>
> > The assumption of “zero-cost” privacy for public information in DClip is overly idealized …
>
> We consider the setting where only the references used to generate text are privacy-sensitive, while the user prompt is not considered privacy-sensitive (similar to the setting considered in previous papers such as [2]). If the reviewer is concerned about the user instructions in the prompt $p$ leaking privacy, this is an interesting question, but it is not the subject of this work.
>
> Note that the use of the previously generated tokens in the public logits does not incur any additional privacy cost due to the postprocessing property of DP (see e.g., Corollary 21 of [1]). Thus, the overall privacy cost (w.r.t. the sensitive references) of generating a full string of T tokens using InvisibleInk is estimated accurately in our Theorem 1.
>
> [1] Steinke. "Composition of differential privacy & privacy amplification by subsampling." Chapter 3 of the book “Differential Privacy in Artificial Intelligence: From Theory to Practice.”
>
> [2] Amin, et al. "Private prediction for large-scale synthetic text generation." EMNLP 2024.
>
> [3] Xie, et al. "Differentially Private Synthetic Data via Foundation Model APIs 2: Text." ICML 2024.
>
> [4] Flemings et al. "Differentially private next-token prediction of large language models." NAACL-HLT 2024.
>
> [5] Flemings et al.. "Adaptively private next-token prediction of large language models." (2024).
>
> [6] Tang et al. "Privacy-preserving in-context learning with differentially private few-shot generation." ICLR 2024.
>
> [7] Mattern, et al. "Differentially private language models for secure data sharing." EMNLP 2022.
>
> [8] Yue et al. "Synthetic text generation with differential privacy: A simple and practical recipe." ACL 2023.
>
> [9] Yuet al. "Privacy-preserving instructions for aligning large language models." ICML 2024.
>
> [10] Tramèr et al.. "Position: Considerations for differentially private learning with large-scale public pretraining." ICML 2024.

---

> > ### Comment · Reviewer_8mTj · 2025-08-05
> >
> > Thank you for your rebuttal. It addressed most of my concerns. I have updated my score to a 4.

---

> > > ### Author Response · Authors · 2025-08-05
> > > **Thank you for the response!**
> > >
> > > Thank you for your response. We are glad to see that the rebuttal addressed most of your concerns.
> > >
> > > We will add the discussions around the whitebox setting of our method and feature the rationale behind the choice of datasets more prominently.
> > >
> > > We would also be happy to address any further questions or concerns you may have. Thanks again!

---

### Official Review · Reviewer_8Xcz · 2025-07-02

**Clarity:** 4
**Significance:** 3
**Originality:** 3
**Rating:** 5
**Confidence:** 4

**Summary:**

This paper proposes a method, InvisibleInk, for privately generating synthetic text data. Unlike other recent approaches such as Private Evolution, which only allows the mechanism API access to the language model, this paper considers the white-box setting, where the mechanism has access to the logits over tokens with respect to next token prediction. This work builds of the method proposed in Amin et al. (2024).

The private component of InvisibleInk uses the exponential mechanism using batch-aggregated logits as the score function with candidate set a superset of the top-k tokens. The score function is the sum of two components: the logits from the language model without using sensitive data as context and the clipped difference between the logits on private data and the public logits. This has the advantage of reducing to the public logits when the privacy budget diminishes to zero. The score function and candidate set are the primary technical contributions.

Empirically, InvisibleInk yields better quality synthetic data than competitor methods.

**Questions:**

Additional Comments
- I suggest moving the AugPE experiments to the main text in the camera ready version.
- In the related work section, it may be helpful to the reader to give the names of the referenced methods such as Augmented PE rather than just pointing to a reference number. I understand this is difficult with respect to Amin et al. (2024).
- In Section 3, is $B$ defined before it is used? Why not just say that $B = |\mathcal{R}|$? The first place I see $B$ formally defined is Line 205.
- On Line 151, do you mean $B < 50$ instead of $B < O(50)$?
- Line 158: you don't need the comma.
- Regarding related work: one way to think about using pretrained models is that they are incorporating public information to augment the private data generation. Several methods for tabular data generation have explored ways to incorporate public data such as prior data releases: PMW-Pub [0], GEM-Pub [1], JAM-PGM [2], Conditional AIM [3].

[0] Liu et al. Leveraging Public Data for Practical Private Query Release (2021)

[1] Liu et al. Iterative Methods for Private Synthetic Data - Unifying Framework and New Methods (2022)

[2] Fuentes et al. Joint Selection: Adaptively Incorporating Public Information for Private Synthetic Data (2024)

[3] Maddock et al. Leveraging Vertical Public-Private Split for Improved Synthetic Data Generation (2025)

**Ethical Concerns:**

["NO or VERY MINOR ethics concerns only"]

**Final Justification:**

This is a strong paper that makes a clear contribution. After the rebuttal, I am keeping my score at an Accept.

**Limitations:**

yes

**Quality:**

4

**Strengths And Weaknesses:**

Strengths
- The proposed method is interesting and the primary contributions directly address weaknesses of existing methods, e.g., Amin et al. (2024).
- The privacy analysis appears correct.
- The empirical evaluation supports the main claims of the paper.
- The paper is well-written and readable.

Weaknesses
- Much of the narrative of the paper is built around the comparison between the proposed method and Amin et al. (2024). AdaPMixED somewhat comes out of nowhere. Perhaps, discussion of AdaPMixED could be better integrated.
- The discussion of the privacy guarantees of prior work is unclear (Lines 255-260).

---

> ### Author Rebuttal · Authors · 2025-07-31
>
> Thank you for the thorough review and the helpful suggestions and pointers! We respond to each point in detail below.
>
> > ... the discussion of AdaPMixED could be better integrated.
>
> This is a good point. We would be happy to elaborate and improve the discussion on AdaPMixED in the main paper.
>
> > The discussion of the privacy guarantees of prior work is unclear (Lines 255-260).
>
> In lines 255-260, we discuss the specifics of the adaptive DP guarantees of prior work like AdaPMixED, and the method proposed by Amin et al. (2024).
>
> Such DP guarantees can only be computed post-generation. For example, the latter method computes the DP guarantees to scale with the number of private tokens selected during SVT. This cannot be known before generation, making it challenging to tune the hyperparameters without a grid search. We use the following heuristic: for a fixed privacy budget and maximum allowed tokens, we set the number of allowed private tokens to be ~25% of all tokens. This heuristic is motivated by empirical observations: in our early explorations of the method of Amin et al. (2024), we found that their SVT step consistently selected approx. 25% of tokens privately, across privacy budgets.
>
> In general, adaptive data-dependent DP guarantees can make precise pre-hoc privacy accounting difficult, as it is infeasible to calculate how many tokens are required to use up the full privacy budget. In contrast, InvisibleInk gives standard DP guarantees and allows pre-hoc privacy accounting. It is straightforward to calculate the maximum number of allowed tokens before generation using the privacy budget and relevant hyperparameters of generation.
>
> For a more detailed discussion on the privacy guarantees of prior work, we refer to Appendix B3 (pp. 25-27).
>
> > In Section 3, B is defined ...
>
> Once again, we thank the reviewer for pointing this out. We are happy to clarify and formally define B along with its first occurrence in Line 112.
>
> > ... Several methods for tabular data generation have explored ways to incorporate public data such as prior data releases...
>
> We thank the reviewer for pointing out additional relevant works.
>
> We note that PMW-Pub, GEM-Pub, and JAM-PGM may be considered analogous to our setting as they use public data to augment private tabular data generation. Further, similar to our experiments on MIMIC (out-of-distribution for pretrained LLMs) and YELP (significant overlap with pretraining data), multiple settings where public data has varying distributional similarity to the private data are studied. Conditional AIM, on the other hand, considers a setting where there exists a vertical split between public and private data; i.e., some attributes (columns) of the dataset are public whereas others are private. This does not have a straightforward analogue in synthetic text generation. A setting where only specific parts (e.g, keywords, proper nouns, etc.) of the reference texts are privacy-sensitive may be an interesting research direction.
>
> We would be happy to include this discussion in the next revision of our work.
>
> > Other minor corrections
>
> We thank the reviewer for the suggestions. We will incorporate them in the next revision of the paper and shift AugPE experiments to the main paper as well.

---

> > ### Author Response · Authors · 2025-08-07
> > **Thank you for your review! Could you please check the rebuttal?**
> >
> > Thank you again for the review! As the discussion period draws to a close, we request you to take a moment to consider our response. We would be happy to elaborate on or clarify any points. Thank you very much!

---

> > ### Comment · Reviewer_8Xcz · 2025-08-08
> >
> > I thank the authors for their detailed rebuttal that cleared up some questions. I'm keeping my score at a 5.

---

> ### Author Response · Authors · 2025-08-08
> **Thank you!**
>
> We are happy to see that the rebuttal has clarified your questions. Thank you once again for your suggestions and for your participation in this discussion!

---

### Official Review · Reviewer_9N2Q · 2025-07-02

**Clarity:** 4
**Significance:** 3
**Originality:** 2
**Rating:** 5
**Confidence:** 4

**Summary:**

There is a strong desire to understand how to add sensitive data to LLM context for the purposes of text generation without having the model leak private information. Differential privacy is a formal privacy guarantee that can ensure that sensitive data are not leaked, but designing DP algorithms for LLM text generation with private context is technically challenging and can be computationally expensive. This work develops several techniques for improving white-box private LLM text generation and combines these techniques into the proposed method InvisibleInk. These techniques include DClip for bounding the contribution of each sensitive text sample tightly and top-k+ for clipping irrelevant tokens from the sampling distribution. Both these methods use public logits (generated from the LLM without any private context) without utilizing the sparse vector technique (SVT), this allows for InvisibleInk to decay gracefully when provided with a small privacy budget as opposed to prior work that can fail catastrophically in the high-privacy regime. The authors demonstrate experimentally that InvisibleInk outperforms prior work both across several metrics and datasets for text generation.

**Questions:**

How would you adapt InvisibleInk to a RAG question answering setting where you need to balance the computation cost and effective temperature tau with the added consideration of retrieving enough information to answer the questions?

You utilize the public logits heavily, how would you handle a setting where the public logits are very different from the private logits (perhaps a very specialized setting where the model needs important context to generate high-quality responses)? Would this change the best value of the clipping parameter?

**Ethical Concerns:**

["NO or VERY MINOR ethics concerns only"]

**Final Justification:**

The reviewers gave helpful context in their responses to my review. This helped give me confidence that my original score of 5 was and still is appropriate. While the paper is a meaningful contribution to the private text generation literature, I do not believe it meet the threshold of being technically flawless and groundbreaking for a 6.

**Limitations:**

The authors discuss the limitations of their methods in the supplementary material. I do not have additional limitations to add.

**Paper Formatting Concerns:**

I do not have any concerns about the formatting of the paper

**Quality:**

4

**Strengths And Weaknesses:**

Strengths:

The work clearly justifies their method from a privacy perspective and also from an LLM decoding perspective. This will allow readers from either background to understand the contributions and why InvisibleInk is effective

The work intelligently incorporates the public logits into DClip and top-k+ to get performance gains from it without using the SVT, which is often unreliable in practice.

InvisibleInk is a user-friendly and practical approach because of the graceful degradation w.r.t. the privacy budget and the useful heuristics for selecting hyperparameters. Hyperparameter tuning under DP is difficult (so many people tune hyperparameters non-privately) but important; the fact that this method comes with practical recommendations for how to set hyperparameters based on the generation length and privacy budget is very useful.

The experiments show general results about the quality of InvisibleInk and they also investigate DClip and top-k+ to demonstrate that these techniques are doing what the authors claim.

Weaknesses:

The authors discuss the batch size B as a quantity that should always be minimized in order to reduce computational cost and increase the number of synthetic records you can generate. However, if you are generating text to answer questions in a RAG system you may need to use more records to gather sufficient context for the generated answer. The introduction mentions this kind of setting but the methods and experiments don’t engage with this setting.

---

> ### Author Rebuttal · Authors · 2025-07-31
>
> Thank you for the thorough review and the thought-provoking questions! We respond to each point in detail below.
>
> > ... how would you handle a setting where the public logits are very different from the private logits ...
>
> In a setting where the public logits are very different from private logits due to very specialized queries, the clipping parameter would have to be increased slightly to preserve downstream utility and quality of the generated text. This results in a larger privacy budget being used up per token generated. (This is unsurprising, as the public logits are now less informative.)
>
> In order to account for such settings, we select 2 out-of-distribution datasets (relative to the pretraining distribution): MIMIC and TAB, for our experiments (MIMIC is commonly used for cross-domain transfer experiments, see e.g., Khaled et al. (2025) [1]). Our empirical results clearly show that InvisibleInk retains much of its performance even at very low privacy budgets in these settings. The loss in utility is negligible in this case, demonstrating that InvisibleInk is sufficiently robust to specialized generation contexts.
>
> > ... in a RAG system you may need to use more records to gather sufficient context ...
>
> This is a great point. It is true that in some practically relevant cases, the batch size $B$ is not necessarily a quantity to be minimized. However, for the previous state-of-the-art methods, the values of $B$ required to generate high-quality text were prohibitively large (ranging from 200-1000). This is in stark contrast to RAG methods that usually retrieve many fewer relevant documents (typically 3-10). In our work, we remove this restriction on large $B$ and allow private text generation to be done in a compute-friendly way for the first time.
>
> RAG is one such potential application of InvisibleInk. In our synthetic text generation setting, batches of sensitive references are sampled (from a partition) from the dataset, agnostic to the query. For RAG, one would have to privatize the selection of the batch of sensitive references, and as the reviewer correctly points out, possibly use more records for sufficient context. The former is a non-trivial and nuanced extension of our work; for the latter, it may suffice to set B to be a reasonable upper bound on the number of references.
>
> However, InvisibleInk is the first such scalable private text generation framework that can act as a starting point for future research on private RAG systems.
>
> > How would you adapt InvisibleInk to a RAG question answering setting ...
>
> Historically, the core task of RAG has been *short-form* question answering. For instance,private RAG methods, such as Koga et al. (2024) [2], focus on *extremely short-form* question answering with output generally limited to 20-30 tokens for a privacy budget of $\varepsilon=10$. InvisibleInk is designed keeping in mind the increasing need for private RAG systems where the output is long-form text (e.g. in recent AI-based search engines such as Perplexity or Google search’s AI-mode).
>
> Our preliminary experiments and ablation studies show that the quality of generation is highly-sensitive to the effective temperature ($\tau$). As previously discussed, RAG places additional constraints on batching as it is no longer agnostic to the query. We hypothesize that the clipping parameter will play an important role in balancing privacy and utility. We conjecture that the improved clipping function, DClip, will thus be crucial for future work in this direction. However, careful experimental evaluation is necessary to understand the nuances of private RAG fully, and is beyond the scope of this work.
>
> [1] Khaled, et al. "Leveraging MIMIC Datasets for Better Digital Health: A Review on Open Problems, Progress Highlights, and Future Promises." arXiv preprint arXiv:2506.12808 (2025).
>
> [2] Koga, et al. "Privacy-Preserving Retrieval-Augmented Generation with Differential Privacy." arXiv preprint arXiv:2412.04697 (2024).

---

### Decision · Program_Chairs · 2025-09-17

**Decision:**

Accept (poster)

**Comment:**

The paper proposed a new approach for private-RAG-based synthetic text generation.  The authors did a good job in the rebuttal in clarifying matters. All concerns are addressed and all four reviewers endorse the paper.  I briefly checked the algorithm and the proof and it appears to be correct.

I'd be happy to recommend accept.